# Enhanced fusogenicity and pathogenicity of SARS-CoV-2 Delta P681R mutation

Akatsuki Saito[1,2,3,22], Takashi Irie[4,22], Rigel Suzuki[5,22], Tadashi Maemura[6,7,22], Hesham Nasser[8,9,22], Keiya Uriu[10,22], Yusuke Kosugi[10,22], Kotaro Shirakawa[11], Kenji Sadamasu[12], Izumi Kimura[10], Jumpei Ito[10], Jiaqi Wu[13,14], Kiyoko Iwatsuki-Horimoto[6], Mutsumi Ito[6], Seiya Yamayoshi[6,15], Samantha Loeber[16], Masumi Tsuda[17,18], Lei Wang[17,18], Seiya Ozono[19], Erika P. Butlertanaka[1], Yuri L. Tanaka[1], Ryo Shimizu[8,20], Kenta Shimizu[5], Kumiko Yoshimatsu[21], Ryoko Kawabata[4], Takemasa Sakaguchi[4], Kenzo Tokunaga[19], Isao Yoshida[12], Hiroyuki Asakura[12], Mami Nagashima[12], Yasuhiro Kazuma[11], Ryosuke Nomura[11], Yoshihito Horisawa[11], Kazuhisa Yoshimura[12], Akifumi Takaori-Kondo[11], Masaki Imai[6,15], The Genotype to Phenotype Japan (G2P-Japan) Consortium*, Shinya Tanaka[17,18 ✉], So Nakagawa[13,14 ✉], Terumasa Ikeda[8 ✉], Takasuke Fukuhara[5 ✉], Yoshihiro Kawaoka[6,7,15 ✉] & Kei Sato[10,14 ✉]

During the current coronavirus disease 2019 (COVID-19) pandemic, a variety of mutations have accumulated in the viral genome of severe acute respiratory syndrome coronavirus 2 (SARS-CoV-2) and, at the time of writing, four variants of concern are considered to be potentially hazardous to human society[1]. The recently emerged B.1.617.2/Delta variant of concern is closely associated with the COVID-19 surge that occurred in India in the spring of 2021 (ref. [2]). However, the virological properties of B.1.617.2/Delta remain unclear. Here we show that the B.1.617.2/Delta variant is highly fusogenic and notably more pathogenic than prototypic SARS-CoV-2 in infected hamsters. The P681R mutation in the spike protein, which is highly conserved in this lineage, facilitates cleavage of the spike protein and enhances viral fusogenicity. Moreover, we demonstrate that the P681R-bearing virus exhibits higher pathogenicity compared with its parental virus. Our data suggest that the P681R mutation is a hallmark of the virological phenotype of the B.1.617.2/Delta variant and is associated with enhanced pathogenicity.

During the current pandemic, SARS-CoV-2 has acquired a variety of mutations[3]. First, in spring 2020, a SARS-CoV-2 derivative containing a D614G mutation in its spike (S) protein emerged and quickly became predominant[4]. As the D614G mutation increases viral infectivity, fitness and interindividual transmissibility[5–10], the D614G-bearing variant quickly outcompeted the original strain. Since autumn 2020, some SARS-CoV-2 variants bearing multiple mutations have emerged and spread rapidly worldwide. As of September 2021, four variants of concern (VOCs) had emerged: B.1.1.7 (Alpha), B.1.351 (Beta), P.1 (Gamma) and B.1.617.2 (Delta)[11,12].

The B.1.617 lineage emerged in India at the end of 2020 and is thought to have been a major driver of the massive COVID-19 surge in India that peaked at 400,000 infection cases per day[2]. The B.1.617 lineage includes three sublineages—B.1.617.1, B.1.617.2 and B.1.617.3. Sublineage B.1.617.2 was defined as the latest VOC as of 25 November 2021, the Delta variant[11,12]. Importantly, early evidence has suggested that infection with B.1.617.2/Delta may carry an increased risk of hospitalization compared with infection with B.1.1.7 (refs. [13–15]). However, the virological features of this newly emerging VOC, particularly its infectivity and pathogenicity, remain unclear. In this study, we demonstrate that B.1.617.2/Delta is more pathogenic than the prototypic SARS-CoV-2 in a Syrian hamster model. We also show that the P681R mutation in the S protein is a hallmark mutation of this lineage. The P681R mutation enhances the

[1]Department of Veterinary Science, Faculty of Agriculture, University of Miyazaki, Miyazaki, Japan. [2]Center for Animal Disease Control, University of Miyazaki, Miyazaki, Japan. [3]Graduate School of Medicine and Veterinary Medicine, University of Miyazaki, Miyazaki, Japan. [4]Institute of Biomedical and Health Sciences, Hiroshima University, Hiroshima, Japan. [5]Department of Microbiology and Immunology, Graduate School of Medicine, Hokkaido University, Hokkaido, Japan. [6]Division of Virology, Institute of Medical Science, University of Tokyo, Tokyo, Japan. [7]Influenza Research Institute, Department of Pathobiological Sciences, School of Veterinary Medicine, University of Wisconsin-Madison, Madison, WI, USA. [8]Division of Molecular Virology and Genetics, Joint Research Center for Human Retrovirus infection, Kumamoto University, Kumamoto, Japan. [9]Department of Clinical Pathology, Faculty of Medicine, Suez Canal University, Ismailia, Egypt. [10]Division of Systems Virology, Department of Infectious Disease Control, International Research Center for Infectious Diseases, The Institute of Medical Science, The University of Tokyo, Tokyo, Japan. [11]Department of Hematology and Oncology, Graduate School of Medicine, Kyoto University, Kyoto, Japan. [12]Tokyo Metropolitan Institute of Public Health, Tokyo, Japan. [13]Department of Molecular Life Science, Tokai University School of Medicine, Kanagawa, Japan. [14]CREST, Japan Science and Technology Agency, Saitama, Japan. [15]The Research Center for Global Viral Diseases, National Center for Global Health and Medicine Research Institute, Tokyo, Japan. [16]Department of Surgical Sciences, School of Veterinary Medicine, University of Wisconsin, Madison, WI, USA. [17]Department of Cancer Pathology, Faculty of Medicine, Hokkaido University, Hokkaido, Japan. [18]Institute for Chemical Reaction Design and Discovery (WPI-ICReDD), Hokkaido University, Hokkaido, Japan. [19]Department of Pathology, National Institute of Infectious Diseases, Tokyo, Japan. [20]Graduate School of Medical Sciences, Kumamoto University, Kumamoto, Japan. [21]Institute for Genetic Medicine, Hokkaido University, Hokkaido, Japan. [22]These authors contributed equally: Akatsuki Saito, Takashi Irie, Rigel Suzuki, Tadashi Maemura, Hesham Nasser, Keiya Uriu, Yusuke Kosugi. *A list of authors and their affiliations appears at the end of the paper. ✉e-mail: tanaka@med.hokudai.ac.jp; so@tokai.ac.jp; ikedat@kumamoto-u.ac.jp; fukut@pop.med.hokudai.ac.jp; yoshihiro.kawaoka@wisc.edu; KeiSato@g.ecc.u-tokyo.ac.jp

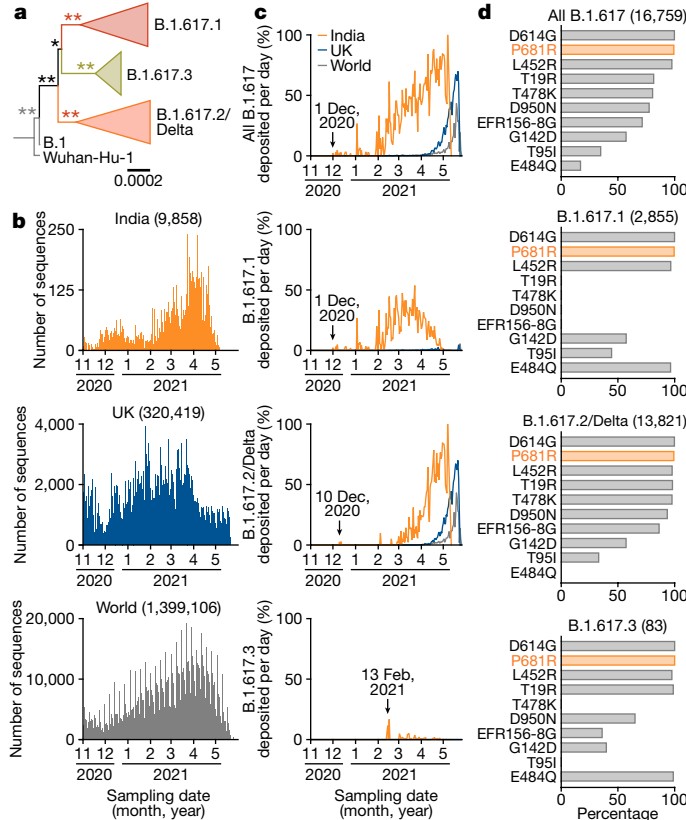

**Fig. 1 | Molecular phylogenetics and epidemic dynamics of the B.1.617 lineage pandemic. a**, Phylogenetic tree of the B.1.617 lineage. Scale bar, 0.0002 substitutions per site. Bootstrap values are indicated by asterisks; **100%, *>70%. The uncollapsed tree is shown in Extended Data Fig. 1. **b**, **c**, Epidemic dynamics of the B.1.617 lineage. **b**, The number of sequences deposited in GISAID per day for India (top), the UK (middle) and the world (bottom). **c**, The percentages of each lineage deposited per day from India (orange), the UK (blue) and the world (grey). The date on which each variant was first identified is indicated. The raw data are summarized in Supplementary Table 1. **d**, The proportion of amino acid replacements in the B.1.617 lineage. The top 10 replacements conserved in the S protein of B.1.617 and its sublineages are summarized. The numbers in parentheses indicate the number of sequences included in each panel. The raw data are summarized in Supplementary Table 2.

cleavage of the SARS-CoV-2 S protein and enhances viral fusogenicity. Moreover, we demonstrate that the P681R mutation can partly explain the higher pathogenicity of the B.1.617.2/Delta variant in vivo.

## Epidemic dynamics of the B.1.617 lineage

We set out to investigate the phylogenetic relationships of the three subvariants belonging to the B.1.617 lineage. We downloaded 1,761,037 SARS-CoV-2 genomes and corresponding data from the Global Initiative on Sharing All Influenza Data (GISAID) database (https://www.gisaid.org; as of 31 May 2021). As expected, each of the three sublineages B.1.617.1, B.1.617.2 and B.1.617.3 formed a monophyletic cluster (Fig. 1a and Extended Data Fig. 1). We next analysed the epidemic dynamics of each of the three B.1.617 sublineages. The B.1.617 variant, specifically B.1.617.1, was first detected in India on 1 December 2020 (GISAID ID: EPI_ISL_1372093) (Fig. 1b, c).

B.1.617.2 (GISAID ID: EPI_ISL_2131509) and B.1.617.3 (GISAID IDs: EPI_ISL_1703672, EPI_ISL_1703659 and EPI_ISL_1704392) were detected in India on 10 December 2020 and 13 February 2021, respectively (Fig. 1c). The prevalence of the B.1.617.1 sublineage peaked from February to April

2021 in India and then decreased (Fig. 1c). Although the B.1.617.3 variant has been detected sporadically in India, the B.1.617.2/Delta lineage has been dominant in India since March 2021 and has also spread all over the world (Fig. 1c). At the end of May 2021, 100%, 70% and 43.3% of the sequences deposited in GISAID per day from India (May 7), the UK (May 21) and the world (May 19), respectively, were B.1.617.2 sublineage sequences (Fig. 1c and Supplementary Table 1).

We next investigated the proportion of amino acid replacements in the S protein of each B.1.617 sublineage compared with the reference strain (Wuhan-Hu-1; GenBank: NC_045512.2). As shown in Fig. 1d, the L452R and P681R mutations were highly conserved in the B.1.617 lineage and, notably, the P681R mutation (16,650 out of 16,759 sequences, 99.3%) was the most representative mutation in this lineage. These data suggest that the P681R mutation is a hallmark of the B.1.617 lineage.

## Syncytium formation by the Delta variant

To investigate the virological characteristics of the B.1.617.2/Delta variant, we conducted virological experiments using an isolate of B.1.617.2 (GISAID ID: EPI_ISL_2378732) as well as a D614G-bearing B.1.1 isolate (GISAID ID: EPI_ISL_479681) in Japan. In Vero cells, the growth of the B.1.617.2/Delta variant was significantly lower compared with the growth of the B.1.1 isolate (Fig. 2a). In particular, the viral RNA levels of the B.1.617.2/Delta variant at 48 h post-infection (h.p.i.) were more than 150-fold lower than those of the B.1.1 isolate (Fig. 2a). By contrast, although the growth kinetics of these viruses were relatively comparable in VeroE6/TMPRSS2 cells and Calu-3 cells (Fig. 2a), microscopy observations showed that the B.1.617.2/Delta variant formed larger syncytia than the B.1.1 virus (Fig. 2b). Measurements of the sizes of the floating syncytia in the infected VeroE6/TMPRSS2 culture indicated that the syncytia stimulated by B.1.617.2/Delta infection were significantly (3.6-fold) larger than those stimulated by B.1.1 infection (Fig. 2b). Moreover, the plaque size in VeroE6/TMPRSS2 cells infected with B.1.617.2/Delta was significantly larger (1.2-fold) compared with in VeroE6/TMPRSS2 cells infected with B.1.1 virus (Extended Data Fig. 2a). Immunofluorescence assays further showed that B.1.617.2/Delta-infected VeroE6/TMPRSS2 cells exhibited larger multinuclear syncytia compared with B.1.1-infected cells (Extended Data Fig. 3a). Notably, although the B.1.1.7/Alpha and B.1.351/Beta VOCs also formed larger syncytia compared with B.1.1, the syncytia formed by B.1.617.2/Delta infection were 1.6-fold and 1.8-fold larger than those formed by B.1.1.7/Alpha and B.1.351/Beta infections, respectively, with statistical significance (Fig. 2b). To directly assess the fusogenicity of the S proteins of these variants, we performed a cell-based fusion assay. We verified that this assay requires expression of human ACE2 in the target cells (Extended Data Fig. 4a). Although the fusogenicity of S proteins of all VOCs tested was significantly greater than that of the parental D614G S, the B.1.617.2/Delta S exhibited the highest fusogenicity with statistical significance (Extended Data Fig. 4b). These results suggest that the B.1.617.2/Delta variant promotes syncytium formation more strongly than the D614G-bearing B.1.1 virus as well as the B.1.1.7/Alpha and B.1.351/Beta VOCs.

## The pathogenicity of the Delta variant

To investigate the pathogenicity of the B.1.617.2/Delta variant, we conducted hamster infection experiments using the B.1.617.2/Delta isolate and the B.1.1 isolate. The viral RNA loads in the oral swabs of B.1.617.2/Delta-infected hamsters were comparable with those of B.1.1-infected hamsters across timepoints on average ($P = 0.057$, multiple regression) (Fig. 2c). Infected hamsters of both groups lost significant body weight beginning at 2 days post-infection (d.p.i.), and the weight loss of B.1.617.2/Delta-infected hamsters was significantly greater than that of B.1.1-infected hamsters across timepoints on average ($P = 0.0082$, multiple regression) (Fig. 2d). The peak weight loss was 16% after infection

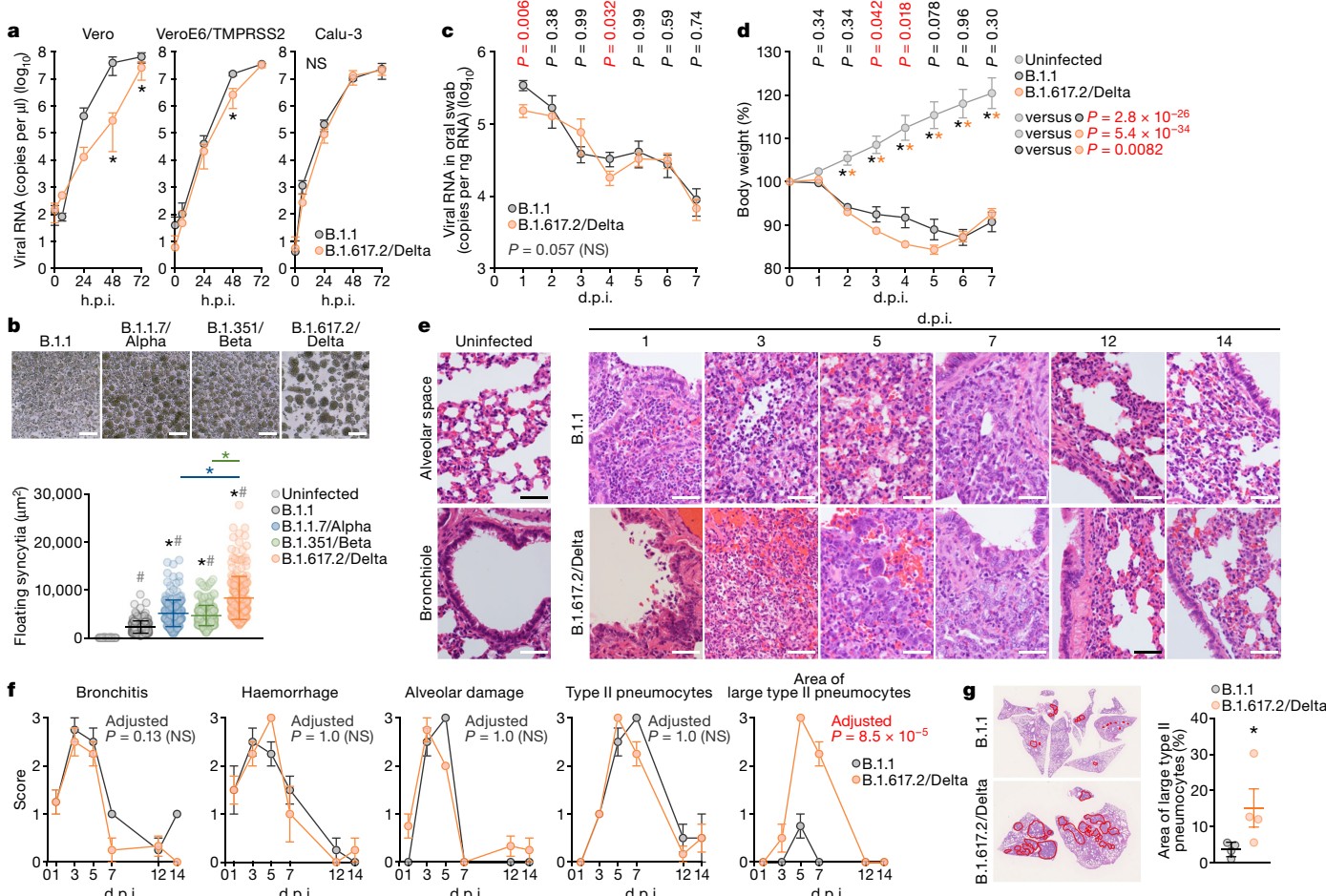

**Fig. 2 | Virological features of the B.1.617.2/Delta variant in vitro and in vivo. a**, Growth kinetics of B.1.617.2/Delta variant. A B.1.617.2/Delta and a D614G-bearing B.1.1 were inoculated in cells, and the copy number of viral RNA in the supernatant was quantified using RT–qPCR. Assays were performed in quadruplicate. **b**, Syncytium formation. Top, representative bright-field images of VeroE6/TMPRSS2 cells at 72 h.p.i. Scale bars, 100 μm. Bottom, the size distributions of floating syncytia in the cultures infected with B.1.1 ($n$ = 215), B.1.1.7/Alpha ($n$ = 199), B.1.351/Beta ($n$ = 249) and B.1.617.2/Delta ($n$ = 216). The size distribution of the floating uninfected cell culture ($n$ = 177) is also shown as a negative control. **c–g**, Infection of Syrian hamsters with the B.1.617.2/Delta variant. Syrian hamsters were intranasally inoculated with B.1.1 ($n$ = 6) and B.1.617.2/Delta ($n$ = 12). Four hamsters of the same age were mock infected. The amount of viral RNA in the oral swab (**c**) and body weight (**d**) were measured. **e**, Haematoxylin and eosin (H&E) staining of the lungs of infected hamsters. Uninfected lung alveolar space and bronchioles are shown (left). Scale bars, 50 μm. **f**, Histopathological scoring of lung lesions. Representative pathological features are shown in Extended Data Fig. 5a. **g**, The area with large

type II pneumocytes in the lungs of B.1.1-infected ($n$ = 4) and B.1.617.2/ Delta-infected ($n$ = 4) hamsters at 5 d.p.i. The area was measured on the photographs (left) and summarized (right, each dot indicates the result from respective hamster). Raw data are shown in Extended Data Fig. 5b. Data are mean ± s.d. (**a**, **b**) or mean ± s.e.m. (**d**, **f**, **g**). In **a**, **b**, **g**, statistically significant differences versus B.1.1, B.1.1.7/Alpha and B.1.351/Beta (*$P$ < 0.05) and uninfected culture ($^{#}P$ < 0.05) were determined using two-sided, unpaired Student's $t$-tests (**a**, **g**) or Mann–Whitney $U$-tests (**b**). In **c**, **d**, **f**, statistically significant differences between B.1.1 and B.1.617.2/Delta were determined by multiple regression and $P$ values (**c**, **d**), and family-wise error rates calculated using the Holm method (**f**) are indicated in the figure. Statistically significant differences at each timepoint were also determined using two-sided unpaired Student's $t$-tests without adjustment for multiple comparisons (**c**, **d**), and those versus uninfected hamsters (*$P$ < 0.05) are indicated by asterisks. The $P$ value of the comparison between B.1.1 and B.1.617.2/Delta at each d.p.i. is indicated in the figure. NS, not significant.

with the B.1.617.2/Delta isolate and 13% for the B.1.1 isolate; hamsters infected with the B.1.617.2/Delta isolate had a significantly greater weight loss compared with B.1.1 at 3 and 4 d.p.i. (Fig. 2d).

In the lungs of infected hamsters of both groups, bronchitis with focal inflammatory cell infiltration around bronchi/bronchioles was observed at 1 d.p.i. followed by haemorrhage or congestion at 3 d.p.i. (Fig. 2e, f and Extended Data Fig. 5a). Crushed nuclear debris, suggesting the damage of the alveolar pneumocytes with macrophage infiltration, was observed from 3 to 5 d.p.i., and the area of inflammatory cell infiltration was expanded with time (Fig. 2e, f and Extended Data Fig. 5a). In both cases, type II pneumocytes with an increased nuclear–cytoplasmic ratio appeared at 5 d.p.i. Notably, in the lungs of B.1.617.2/

Delta-infected hamsters, prominently enlarged cells with large nuclei (greater than 10 μm in diameter) were recognized, suggesting large type II pneumocytes that reflect the severity of pneumonia at 5 d.p.i. (Fig. 2f, g and Extended Data Fig. 5b). Immunohistochemistry analysis of viral nucleocapsid (N) protein demonstrated that N proteins were detected in the bronchial epithelial cells with a small fraction of alveolar staining in both infection cases at 1 d.p.i. (Extended Data Fig. 5c). In the case of B.1.1 infection, N proteins were detected equally in bronchi/ bronchioles at 1 and 3 d.p.i. (Extended Data Fig. 5c). At 5 d.p.i., alveolar pneumocytes exhibited positivity for N protein, which was weakened at 7 d.p.i. (Extended Data Fig. 5c). By contrast, in the case of B.1.617.2/Delta infection, the areas that were positive for N protein migrated rapidly to

the alveolar pneumocytes around the bronchi/bronchioles and most of the bronchial epithelium was negative at 3 d.p.i. (Extended Data Fig. 5c). Thereafter, the N-positive areas further moved to the periphery of the lung lobes at 5 d.p.i. and were undetectable at 7 d.p.i. (Extended Data Fig. 5c). These observations suggest that the spaciotemporal distribution of infected cells between B.1.617.2/Delta and B.1.1 are different, and that the B.1.617.2/Delta isolate has higher pathogenicity in terms of the rapid spreading from bronchi/bronchioles to the alveolar space reaching the lung periphery than the B.1.1 isolate in spite of their relatively comparable proliferative potential.

## The effect of the P681R mutation on viral fusion

The P681R mutation in the S protein is a unique feature of the B.1.617 lineage, including the B.1.617.2/Delta variant (Fig. 1d). As the P681R mutation is located in proximity to the furin cleavage site (FCS; residues RRAR positioned at 682–685) of the SARS-CoV-2 S protein[16], we hypothesized that the P681R mutation is responsible for the promotion of cell–cell fusion, leading to the formation of larger syncytia. To address this possibility, we generated a P681R-bearing artificial virus by reverse genetics (Extended Data Fig. 2b) and performed further virological experiments. Although the amounts of viral RNA in the culture supernatants of the D614G/P681R-infected Vero and VeroE6/TMPRSS2 cells were significantly lower compared with those of the D614G-infected cells at some timepoints, the growth of these two viruses was relatively comparable (Fig. 3a). However, the floating syncytia (Fig. 3b) and plaques (Extended Data Fig. 2c) in the D614G/P681R-infected VeroE6/TMPRSS2 cells at 72 h.p.i. were significantly larger in size compared with the syncytia in the D614G-mutant-infected cells. Moreover, immunofluorescence assays showed that D614G/P681R-infected VeroE6/TMPRSS2 cells exhibited larger multinuclear cells than D614G-infected cells (Extended Data Fig. 3b). These observations correspond well to the observations in the culture infected with the B.1.617.2/Delta variant (Fig. 2b and Extended Data Figs. 2a, 3a).

To clearly observe syncytium formation, we further generated GFP-expressing replication-competent D614G and D614G/P681R viruses. The levels of viral RNA in the supernatant and proportion of GFP-positive cells were similar in Vero, VeroE6/TMPRSS2 and Calu-3 cells (Extended Data Fig. 6). However, at 24 h.p.i., significantly larger GFP-positive adherent syncytia were observed in VeroE6/TMPRSS2 cells infected with the GFP-expressing D614G/P681R virus (Fig. 3c). Moreover, the GFP-positive floating syncytia at 72 h.p.i. in VeroE6/TMPRSS2 cells infected with GFP-expressing D614G/P681R virus were significantly larger (2.4-fold) in size compared with those of VeroE6/TMPRSS2 cells infected with GFP-expressing D614G virus (Extended Data Fig. 7a). Moreover, GFP-positive syncytia were observed in D614G/P681R-infected Calu-3 cells but not in D614G-infected Calu-3 cells at 72 h.p.i. (Extended Data Fig. 6c). These results suggest that the features of the B.1.617.2/Delta virus observed in in vitro cell culture experiments, particularly the formation of larger syncytia, are well reproduced by the insertion of the P681R mutation. To further investigate the effect of the P681R mutation, the GFP-expressing viruses were inoculated into human primary nasal epithelial culture. Notably, the viral RNA levels of D614G/P681R virus on the apical side of culture at 2 and 3 d.p.i. were 12.3-fold and 7.0-fold higher, respectively, than those of parental D614G virus with statistical significance, and the rapid growth of D614G/P681R virus was supported by the observation of GFP expression (Extended Data Fig. 7b). Although the viral RNA levels of D614G/P681R virus gradually decreased after 5 d.p.i., plaque-like spots were observed after 7 d.p.i., and the sizes of these plaque-like spots in the culture of D614G/P681R infection were significantly larger than the plaque-like spots in the culture of parental D614G virus infection (Extended Data Fig. 7b). These data suggest that the P681R mutation accelerates viral replication in human primary nasal epithelial culture and produces large plaque-like spots, which could be formed by cell-to-cell infection as the case of plaque formation.

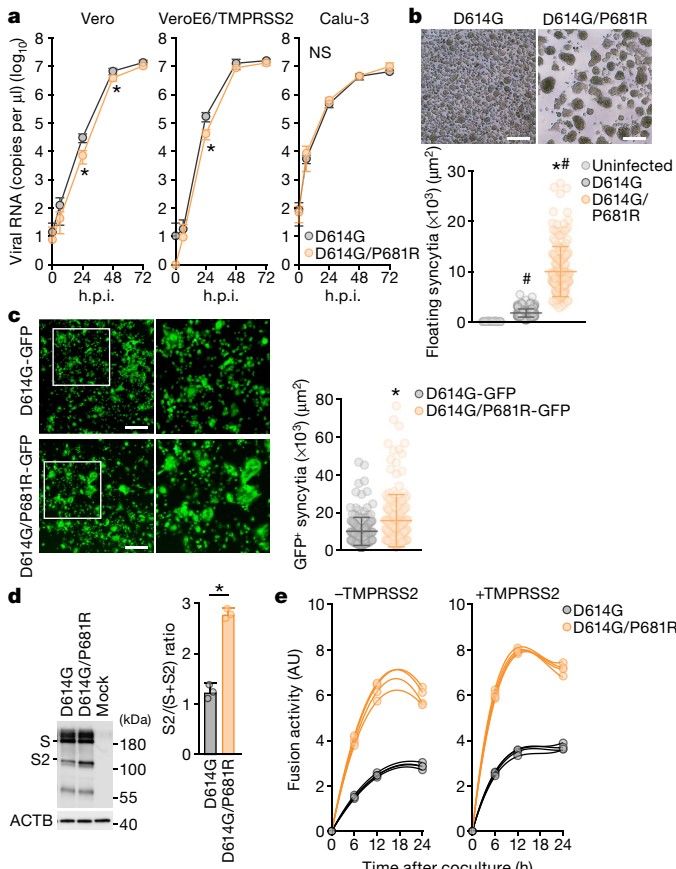

**Fig. 3 | Virological features of the P681R-containing virus in vitro. a**, The growth kinetics of artificially generated viruses. The D614G and D614G/P681R mutant viruses were generated by reverse genetics. These viruses (100 tissue culture infectious dose ($TCID_{50}$)) were inoculated into Vero cells and VeroE6/TMPRSS2 cells, and the copy number of viral RNA in the culture supernatant was quantified using RT–qPCR. The growth curves of the inoculated viruses are shown. Assays were performed in quadruplicate. **b**, **c**, Syncytium formation. **b**, Floating syncytia in VeroE6/TMPRSS2 cells infected with the D614G and D614G/P681R mutant viruses at 72 h.p.i. (top). Scale bars, 200 μm. Bottom, the size distributions of floating syncytia in D614G-infected ($n = 228$) and D614G/P681R-infected ($n = 164$) cultures. **c**, Adherent syncytia in VeroE6/TMPRSS2 cells infected with GFP-expressing D614G- and D614G/P681R-mutant viruses at 24 h.p.i. Higher-magnification views of the regions indicated by with squares are shown in the right images. Scale bars, 200 μm. The size distributions of adherent GFP+ syncytia in the D614G-infected ($n = 111$) and D614G/P681R-infected ($n = 126$) cultures. **d**, Western blot analysis of S-expressing cells. Left, representative blots of SARS-CoV-2 full-length S and cleaved S2 proteins as well as ACTB as an internal control. Assays were performed in triplicate. Data are mean ± s.d. Right, the ratio of S2 to the full-length S plus S2 proteins in the S-expressing cells. **e**, SARS-CoV-2 S-based fusion assay. Effector cells (S-expressing cells) and target cells (ACE2-expressing cells or ACE2/TMPRSS2-expressing cells) were prepared, and the fusion activity was measured as described in the Methods. Assays were performed in quadruplicate, and fusion activity (arbitrary units) is shown. Data are mean ± s.d. Statistically significant differences versus D614G (*$P < 0.05$) and uninfected culture (#$P < 0.05$) were determined using two-sided unpaired Student's t-tests (**a**, **d**) or Mann–Whitney U-tests (**b**, **c**).

To directly investigate the effect of the P681R mutation on the cleavage of the SARS-CoV-2 S protein, we prepared an HIV-1-based pseudovirus carrying the P681R mutation. Western blot analysis of the prepared pseudoviruses showed that the level of the cleaved S2 subunit was significantly increased in the presence of the P681R mutation (Extended Data Fig. 8a), suggesting that the P681R mutation

facilitates furin-mediated cleavage of the SARS-CoV-2 S protein. We next performed a single-round pseudovirus infection assay using target HOS-ACE2 cells with or without *TMPRSS2* expression. The infectivity of both the D614G and D614G/P681R pseudoviruses was increased approximately tenfold by the expression of *TMPRSS2* in the target cells (Extended Data Fig. 8b). However, the relative infectivity of the D614G and D614G/P681R pseudoviruses was not altered by *TMPRSS2* expression (Extended Data Fig. 8b). These data suggest that the P681R mutation does not affect the infectivity of the viral particles.

We next addressed the effect of the P681R mutation on viral fusogenicity by a cell-based fusion assay. In the effector cells (that is, S-expressing cells), although the expression level of the D614G/P681R S protein was comparable to that of the D614G S protein, the level of the cleaved S2 subunit was significantly higher for the D614G/P681R mutant than for the D614G mutant (Fig. 3d). Consistent with the results of the pseudovirus assay (Extended Data Fig. 8a), these results suggest that the P681R mutation facilitates S cleavage. Flow cytometry analysis showed that the surface expression level of D614G/P681R S was significantly lower than that of D614G S (Extended Data Fig. 8c). Nevertheless, the cell-based fusion assay using the target cells without TMPRSS2 demonstrated that D614G/P681R S is 2.1-fold more fusogenic than D614G S—a statistically significant difference ($P = 0.0002$, Welch's *t*-test) (Fig. 3e). Moreover, a mathematical modelling analysis of the fusion assay data showed that the initial fusion velocity of D614G/P681R S ($0.83 \pm 0.03$ per hour) was significantly faster (2.8-fold) than that of D614G S ($0.30 \pm 0.03$ per hour; $P = 4.0 \times 10^{-6}$, Welch's *t*-test) (Extended Data Fig. 8d, e). These data suggest that the P681R mutation enhances and accelerates SARS-CoV-2 S-mediated fusion. Furthermore, when we used targeted cells expressing *TMPRSS2*, both the fusion efficacy (about 1.2-fold) and initial fusion velocity (about 2.0-fold) were increased in both the D614G and D614G/P681R S proteins (Extended Data Fig. 8d, e). These results suggest that TMPRSS2 facilitates the fusion mediated by SARS-CoV-2 S and human ACE2 and that this TMPRSS2-dependent acceleration and promotion of viral fusion is not specific for the P681R mutant.

## Neutralization of the P681R mutant

Resistance to neutralizing antibodies in the sera of COVID-19 convalescent individuals and vaccinated individuals is a hallmark of VOCs[17,18], and it has recently been shown that the B.1.617.2/Delta variant is relatively resistant to vaccine-induced neutralization[19,20]. To determine whether the P681R mutation contributes to this virological phenotype, we performed a neutralization assay. The D614G/P681R pseudovirus was partially resistant (1.2–1.5-fold) to three monoclonal antibodies targeting the receptor-binding domain of the SARS-CoV-2 S protein (Extended Data Fig. 9a). Furthermore, neutralization experiments using 19 serum samples collected after two rounds of BNT162b2 vaccination showed that the D614G/P681R pseudovirus was significantly more resistant than the D614G pseudovirus to vaccine-induced neutralizing antibodies ($P < 0.0001$, Wilcoxon matched-pairs signed-rank test) (Extended Data Fig. 9b, c). These results suggest that the P681R-bearing pseudovirus is relatively resistant to neutralizing antibodies.

## Pathogenicity of the P681R mutant

To assess the effect of the P681R mutation on viral replication and the pathogenicity of SARS-CoV-2, we intranasally infected Syrian hamsters with the D614G and D614G/P681R viruses. The D614G-infected hamsters exhibited no weight loss, although a slight decrease in body weight by 7 d.p.i. was observed for one of the hamsters (5.0%) (Fig. 4a). By contrast, all of the hamsters infected with the D614G/P681R virus experienced gradual body weight loss, and the hamsters showed a weight loss of 4.7–6.9% at 7 d.p.i., significantly greater compared with the weight loss of hamsters that were infected with the D614G virus

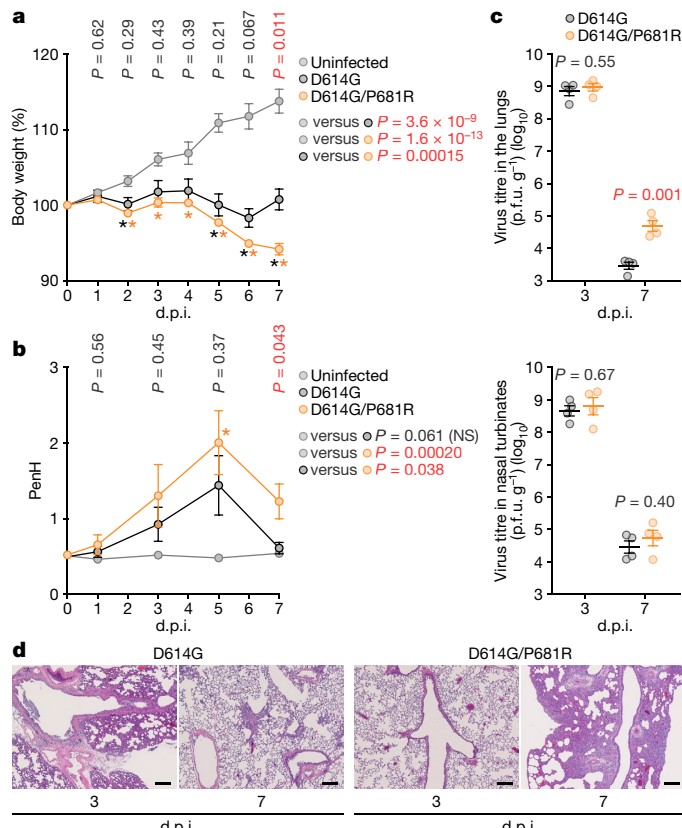

**Fig. 4 | Enhanced pathogenicity by the P681R mutation in hamsters.** Syrian hamsters were intranasally inoculated with the D614G and D614G/P681R viruses. **a**, Body weight changes in hamsters after viral infection. Body weights of virus-infected ($n = 4$ each) and uninfected ($n = 3$) hamsters were monitored daily for 7 days. **b**, Pulmonary function analysis in infected hamsters. Enhanced pause (PenH), which is a surrogate marker for bronchoconstriction or airway obstruction, was measured using whole-body plethysmography. **c**, Virus replication in infected hamsters. Four hamsters per group were euthanized at 3 d.p.i. and 7 d.p.i. for virus titration. Virus titres in the lungs (top) and nasal turbinates (bottom) were determined by plaque assay using VeroE6/TMPRSS2 cells. The points indicate data from individual Syrian hamsters. p.f.u., plaque-forming units. **d**, Histopathological examination of the lungs of infected Syrian hamsters. Representative pathological images of D614G- and D614G/P681R-infected lungs at 3 d.p.i. and 7 d.p.i. Scale bars, 200 μm. Data are mean ± s.e.m. In **a**, **b**, statistically significant differences were determined by multiple regression and *P* values are indicated in the figure. Statistically significant difference at each timepoint was also determined using two-sided unpaired Student's *t*-tests without adjustment for multiple comparisons, and those versus uninfected hamsters (*$P < 0.05$) are indicated by asterisks. The *P* value of the comparison between D614G and D614G/P681R at each d.p.i. is indicated in the figure.

($P = 0.011$) (Fig. 4a). The weight loss of D614G/P681R-infected hamsters was significantly greater compared with that of D614G-infected hamsters on average across all timepoints ($P = 0.00015$, multiple regression) (Fig. 4a). We also assessed pulmonary function in infected hamsters by using a whole-body plethysmography system to measure enhanced pause (PenH), which is a surrogate marker of bronchoconstriction or airway obstruction. Infected hamsters of both groups showed increases in the lung PenH value, but the PenH values of D614G/P681R-infected hamsters were significantly higher than those of the D614G-infected hamsters on average across all timepoints ($P = 0.038$, multiple regression) (Fig. 4b). At 7 d.p.i., the D614G/P681R-infected hamsters had significantly higher PenH values than the D614G-infected hamsters ($P = 0.043$). At 3 d.p.i., both viruses replicated efficiently in the lungs

and nasal turbinates of the infected hamsters, and no significant difference in viral replication was observed between the two groups (Fig. 4c). At 7 d.p.i., no differences in viral titres in the nasal turbinates were found between the two groups; however, the lung titres in the D614G/P681R-infected group were significantly higher than those in the D614G-infected group ($P = 0.0013$) (Fig. 4c).

Histopathological examination revealed cell infiltration in and around the bronchi/bronchioles at 3 d.p.i. in both groups, but solid bronchioloalveolar epithelial hyperplasia including type II pneumocytes was prominent at 7 d.p.i. in the D614G/P681R-infected hamsters (Fig. 4d). Microcomputed tomography (microCT) analysis revealed lung abnormalities in all of the infected hamsters on 7 d.p.i. that were consistent with commonly reported imaging features of COVID-19 pneumonia[21] (Extended Data Fig. 10a). Lung abnormalities included multifocal nodular ground glass opacity with a peripheral, bilateral, multilobar, peribronchial distribution with regions of lung consolidation. The CT severity scores of the D614G-infected and D614G/P681R-infected hamsters ranged from 8 to 14, with an overall average CT severity score of 10.5 (median 9.5) (Extended Data Fig. 10b). The D614G/P681R-infected hamsters had a higher CT severity score (mean 11 (range 9–14, median 10.5)), compared with the D614G-infected hamsters (mean 10 (range 8–13, median 9.5)). Two of the D614G/P681R-infected hamsters developed a small-volume pneumomediastinum, probably secondary to severe pulmonary damage, micropulmonary rupture and gas tracking into the mediastinum.

## Discussion

Previous studies have demonstrated the close association of FCS in the SARS-CoV-2 S protein with the viral replication mode and its dependence on TMPRSS2. Johnson et al.[23] and Peacock et al.[22] showed that the loss of FCS results in an increase in viral replication efficacy in Vero cells and attenuates viral growth in Vero cells expressing *TMPRSS2*. By contrast, here we showed that the replication efficacy of the B.1.617.2/Delta variant was severely decreased in Vero cells compared with VeroE6/TMPRSS2 cells. Importantly, although FCS-deleted SARS-CoV-2 is less pathogenic compared with its parental virus[23], we revealed that the B.1.617.2/Delta variant and the P681R-harbouring virus exhibit higher pathogenicity. These findings suggest that enhanced viral fusogenicity, which is triggered by the P681R mutation, is closely associated with viral pathogenicity.

Although the P681R mutant is highly fusogenic, the virus containing the P681R mutation did not necessarily show stronger growth than the parental virus in in vitro cell cultures. HIV-1 variants with higher fusogenicity have been isolated from patients with AIDS, but the enhanced fusogenicity does not promote viral replication in in vitro cell cultures[24]. Similarly, a measles virus (*Paramyxoviridae*) containing mutations in viral matrix proteins[25] and substitution mutations in viral fusion proteins[26,27] is highly fusogenic and expands efficiently through cell–cell fusion. However, in in vitro cell cultures, the growth kinetics of these mutated measles viruses with higher fusogenicity are less efficient compared with those of the parental virus[25]. Thus, the discrepancy between the efficacy of viral growth in in vitro cell cultures and viral fusogenicity is not unique to SARS-CoV-2. However, higher fusogenicity is associated with the severity of viral pathogenicity, such as in HIV-1 encephalitis[28] and the fatal subacute sclerosing panencephalitis that is caused by measles virus infection in the brain[26,27]. Consistently, we showed that both the B.1.617.2/Delta variant and the P681R mutant exhibited higher fusogenicity in vitro and enhanced pathogenicity in vivo. Our data suggest that the greater COVID-19 severity and unusual symptoms caused by the B.1.617.2/Delta variant[13–15] are due in part to the higher fusogenicity caused by the P681R mutation.

After launching this research in May 2021, the B.1.617.2/Delta variant has rapidly surpassed the other VOCs and is a major driver of the current COVID-19 pandemic worldwide in only a few months after the emergence. Revealing the rationale of higher transmissibility of this variant is one of the most urgent and crucial issues in the current COVID-19 pandemic. However, transmission experiments using the B.1.617.2/Delta variant or P681R-bearing virus in animal models were not performed in this study, and it remains unaddressed why the B.1.617.2/Delta variant has become more predominant than the other VOCs. By contrast, the greater severity and unusual COVID-19 symptoms caused by the B.1.617.2/Delta variant[13–15] should be another important issue; we therefore addressed the virological properties and virulence of this variant and showed evidence suggesting that this pandemic variant has enhanced fusogenicity and pathogenicity. We revealed the association of the P681R mutation with these virological features. An assumption from our observations is that the higher viral fusogenicity driven by the P681R mutation may be associated with the increased transmissibility of the B.1.617.2/Delta variant observed in humans. However, the P681R mutation is not specific for the B.1.617.2/Delta variant, and the sublineages related to the B.1.617.2/Delta variant, such as the B.1.617.1 and B.1.617.3 variants, that contain this mutation have not successfully spread in the human population. Thus, the mutations unique for the B.1.617.2/Delta variant would determine its higher transmissibility and further investigation will be needed to elucidate this property of the B.1.617.2/Delta variant.

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

**The Genotype to Phenotype Japan (G2P-Japan) Consortium**

**Mika Chiba[10], Hirotake Furihata[10], Haruyo Hasebe[8], Kazuko Kitazato[8], Haruko Kubo[5], Naoko Misawa[10], Nanami Morizako[1], Kohei Noda[8], Akiko Oide[10], Mai Suganami[10], Miyoko Takahashi[13], Kana Tsushima[5], Miyabishara Yokoyama[10] & Yue Yuan[8]**

# Methods

## Ethics statement

The virus isolation procedures in this study were approved by the Institutional Review Board of Tokyo Metropolitan Institute of Public Health according to the Declaration of Helsinki 2013 (3KenKenKen-466). All protocols involving specimens from human subjects recruited at Kyoto University were reviewed and approved by the Institutional Review Boards of Kyoto University (G0697) and the Institute of Medical Science, the University of Tokyo (2021-1-0416). All of the human participants provided written informed consent. All of the experiments with hamsters were performed in accordance with the Science Council of Japan's Guidelines for Proper Conduct of Animal Experiments. The protocols were approved by the Institutional Animal Care and Use Committee of National University Corporation Hokkaido University (20-0123) and the Animal Experiment Committee of the Institute of Medical Science, the University of Tokyo (PA19-75).

## Collection of BNT162b2-vaccinated sera

Peripheral blood was collected four weeks after the second vaccination with BNT162b2 (Pfizer-BioNTech), and sera were isolated from the peripheral blood of 19 vaccinees (average age, 38; range, 28–59; 26% male). Sera were inactivated at 56 °C for 30 min and stored at −80 °C until use.

## Cell culture

HEK293 cells (a human embryonic kidney cell line; ATCC CRL-1573), HEK293T cells (a human embryonic kidney cell line; ATCC CRL-3216) and HOS cells (a human osteosarcoma cell line; ATCC CRL-1543) were maintained in Dulbecco's modified Eagle's medium (high glucose) (Wako, 044-29765) containing 10% fetal bovine serum (FBS) and 1% penicillin–streptomycin (PS). Vero cells (an African green monkey (*Chlorocebus sabaeus*) kidney cell line; JCRB0111) were maintained in Eagle's minimum essential medium (Wako, 051-07615) containing 10% FBS and 1% PS. VeroE6/TMPRSS2 cells (an African green monkey (*C. sabaeus*) kidney cell line; JCRB1819)[29] were maintained in Dulbecco's modified Eagle's medium (low glucose) (Wako, 041-29775) containing 10% FBS, G418 (1 mg ml$^{-1}$; Nacalai Tesque, G8168-10ML) and 1% PS. Calu-3 cells (a human lung epithelial cell line; ATCC HTB-55) were maintained in Eagle's minimum essential medium (Sigma-Aldrich, M4655-500ML) containing 10% FCS and 1% PS. HOS-ACE2/TMPRSS2 cells, HOS cells stably expressing human *ACE2* and *TMPRESS2*, were prepared as previously described[30,31]. HEK293-C34 cells, *IFNAR1*-KO HEK293 cells expressing human *ACE2* and *TMPRSS2* by doxycycline treatment[32], were maintained in Dulbecco's modified Eagle's medium (high glucose) (Sigma-Aldrich, R8758-500ML) containing 10% FBS, 10 μg ml$^{-1}$ blasticidin (InvivoGen, ant-bl-1) and 1% PS. Primary human nasal epithelial cells (EP01, MD0436) were purchased from Epithelix and maintained according to the manufacturer's procedure.

## Animal experiments

Syrian hamsters (male, 4 weeks old) were purchased from Japan SLC. Baseline body weights were measured before infection. For the virus infection experiments in Fig. 2c, d, hamsters were euthanized by intramuscular injection of a mixture of 0.15 mg kg$^{-1}$ medetomidine hydrochloride (Domitor, Nippon Zenyaku Kogyo), 2.0 mg kg$^{-1}$ midazolam (Dormicum, Maruishi Pharmaceutical) and 2.5 mg kg$^{-1}$ butorphanol (Vetorphale, Meiji Seika Pharma). The B.1.1 or B.1.167.2/Delta viruses ($10^5$ TCID$_{50}$ in 100 μl) were intranasally infected under anaesthesia. Body weights were measured, and oral swabs were collected under anaesthesia with isoflurane (Sumitomo Dainippon Pharma) daily. For the virus infection in Fig. 4, four hamsters per group were intranasally inoculated with the D614G or the D614G/P681R viruses ($10^4$ TCID$_{50}$ in 30 μl) under isoflurane anaesthesia. Body weight was monitored daily for 7 days. For virological examinations, four hamsters per group were intranasally infected with the D614G or the D614G/P681R viruses

($10^4$ TCID$_{50}$ in 30 μl); at 3 and 7 d.p.i., the hamsters were euthanized, and nasal turbinates and lungs were collected. The virus titres in the nasal turbinates and lungs were determined by plaque assays in VeroE6/TMPRSS2 cells.

## Histopathological analysis

Excised animal tissues were fixed with 4% paraformaldehyde in PBS, and processed for paraffin embedding. The paraffin blocks were sectioned with at a thickness of 3 μm and then mounted on silane-coated glass slides (MAS-GP, Matsunami). H&E staining was performed according to a standard protocol. For immunohistochemistry analysis (Extended Data Fig. 5c), an Autostainer Link 48 (Dako) was used. The deparaffinized sections were exposed to EnVision FLEX target retrieval solution high pH (Agilent, K8004) for 20 min at 97 °C to activate, and a mouse anti-SARS-CoV-2 N monoclonal antibody (1:400 dilution; R & D systems, 1035111, MAB10474-SP) was used. The sections were sensitized using EnVision FLEX (Agilent) for 15 min and visualized by peroxidase-based enzymatic reaction with 3,3′-diaminobenzidine tetrahydrochloride as the substrate for 5 min.

## Histopathological scoring of lung lesions

Pathological features, including bronchitis or bronchiolitis, haemorrhage or congestion, alveolar damage with epithelial apoptosis and macrophage infiltration, the presence of type II pneumocytes and the area of the presence of large type II pneumocytes (Fig. 2f and Extended Data Fig. 5b), were evaluated by certified pathologists and the degree of these pathological findings were arbitrarily scored using four-tiered system as 0 (negative), 1 (weak), 2 (moderate) and 3 (severe). Especially, for the evaluation of the area of the large type II pneumocytes at 5 d.p.i., the presence of more than 5 large type II pneumocytes with a nuclear diameter more than 10 μm per 0.04 mm$^2$ were delineated and the areas were measured using Fiji software v.2.2.0 implemented in ImageJ v.2.2.0.

## Lung function

Respiratory parameters were measured using a whole-body plethysmography system (PrimeBioscience) according to the manufacturer's instructions. In brief, hamsters were placed in unrestrained plethysmography chambers and allowed to acclimatize for 1 min, then data were acquired over a 3 min period using FinePointe v.2.8.0.12146 (Data Sciences International).

## MicroCT imaging

Respiratory organs of the infected hamsters were imaged by using an in vivo microCT scanner (CosmoScan GXII; Rigaku) at 7 d.p.i. Under ketamine–xylazine and isoflurane for the induction and maintenance of anaesthesia, the hamsters were placed in the imaging chamber and were scanned for 4 min at 90 kV, 88 μA, FOV 45 mm and a pixel size 90.0 μm. After scanning, the lung images were reconstructed and analysed using the CosmoScan Database software v.3.3.27.100 (Rigaku).

Qualitative and semiquantitative visual image analysis of the lungs was performed in three uninfected Syrian hamsters and the hamsters infected with D614G ($n = 4$) or D614G/P681R ($n = 4$) viruses at 7 d.p.i. A CT severity score (Extended Data Fig. 10b), which was adapted from a human scoring system, was used to grade the severity of the lung abnormalities[33]. Each lung lobe was analysed for degree of involvement and was scored from 0–4 as follows depending on the severity: 0 (none, 0%), 1 (minimal, 1%–25%), 2 (mild, 26%–50%), 3 (moderate, 51%–75%) or 4 (severe, 76%–100%). The scores for the five lung lobes were summed to obtain a total severity score of 0–20, reflecting the severity of abnormalities across the two infected groups. Images were anonymized and randomized; the scorer was blinded to the group allocation.

## Viral genomes

All SARS-CoV-2 genome sequences and annotation information used in this study were downloaded from GISAID (https://www.gisaid.org)

on 31 May 2021 (1,761,037 sequences). We first excluded genomes of viruses collected from non-human hosts. We obtained SARS-CoV-2 variants belonging to the B.1.617 lineage based on the PANGO annotation (that is, sublineages B.1.617.1, B.1.617.2/Delta or B.1.617.3) for each sequence in the GISAID metadata. One variant annotated as belonging to the B.1.617 lineage (GISAID ID: EPI_ISL_1544002, isolated in India on 25 February 2021) was not used in the analysis because the variant was not assigned to any of the three sublineages, possibly due to the 212 undetermined nucleotides in the genome. To infer the epidemiology of the B.1.617 lineage (Fig. 1b–d), we excluded genomes for which sampling date information was not available. We analysed 2,855, 13,821 and 83 sequences belonging to the B.1.617.1, B.1.617.2/Delta and B.1.617.3 sublineages, respectively.

A SARS-CoV-2 variant (GISAID ID: EPI_ISL_2220643) isolated in Texas, USA, on 10 August 2020, was also recorded to belong to B.1.617.1. However, the S protein of this viral sequence (GISAID ID: EPI_ISL_2220643) possesses neither L452R nor P681R mutations, both of which are features of the B.1.617 lineage. Thus, the EPI_ISL_2220643 sequence isolated in the USA is probably not the ancestor of the current B.1.617.1 lineage, and the EPI_ISL_1372093 sequence obtained in India can be considered to be the oldest example of the B.1.617 lineage.

### Phylogenetic analyses
To infer the phylogeny of the B.1.617 sublineages, we screened SARS-CoV-2 genomes by removing genomes containing undetermined nucleotides at coding regions. As the numbers of genomes belonging to sublineages B.1.617.1 and B.1.617.2/Delta are large (894 and 6152 sequences, respectively), we used 150 randomly chosen sequences for each sublineage. For the B.1.617.3 sublineage, 32 genomes were used. We used the Wuhan-Hu-1 strain isolated in China on 31 December 2019 (GenBank: NC_045512.2 and GISAID ID: EPI_ISL_402125) and the LOM-ASST-CDG1 strain isolated in Italy on 20 February 2020 (GISAID ID: EPI_ISL_412973) together as an outgroup. We next collected 334 representative SARS-CoV-2 sequences and aligned the entire genome sequences using the FFT-NS-1 program in the MAFFT suite (v.7.407)[34]. All sites with gaps in the alignment were removed, and the total length of the alignment was 29,085 nucleotides. A maximum likelihood tree was generated using IQ-TREE 2 v.2.1.3 with 1,000 bootstraps[35]. The GTR+G substitution model was used based on the BIC criterion.

### SARS-CoV-2 preparation and titration
A B.1.617.2/Delta isolate (strain TKYTK1734; GISAID ID: EPI_ISL_2378732) and a D614G-bearing B.1.1 isolate (strain TKYE610670; GISAID ID: EPI_ISL_479681) were isolated from SARS-CoV-2-positive individuals in Japan. In brief, 100 µl of nasopharyngeal swabs obtained from SARS-CoV-2-positive individuals were inoculated into VeroE6/TMPRSS2 cells in a biosafety level 3 laboratory. After incubation at 37 °C for 15 min, maintenance medium supplemented with Eagle's minimum essential medium (FUJIFILM Wako Pure Chemical, 056-08385) containing 2% FBS and 1% PS was added, and the cells were cultured at 37 °C under 5% $CO_2$. The cytopathic effect (CPE) was confirmed by observation under an inverted microscope (Nikon), and the viral load of the culture supernatant in which CPE was observed was confirmed by RT–qPCR. To determine the viral genome sequences, RNA was extracted from the culture supernatant using the QIAamp viral RNA mini kit (Qiagen, 52906). A cDNA library was prepared using the NEB Next Ultra RNA Library Prep Kit for Illumina (New England Biolab, E7530) and whole-genome sequencing was performed by MiSeq (Illumina).

To prepare the working virus stock, 100 µl of the seed virus was inoculated into VeroE6/TMPRSS2 cells ($5 \times 10^6$ cells in a T-75 flask). Then, 1 h after infection, the culture medium was replaced with Dulbecco's modified Eagle's medium (low glucose) (Wako, 041-29775) containing 2% FBS and 1% PS. At 2–3 d.p.i., the culture medium was collected and centrifuged, and the supernatants were collected as the working virus stock.

The titre of the prepared working virus was measured as the 50% tissue culture infectious dose ($TCID_{50}$). In brief, 1 day before infection, VeroE6/TMPRSS2 cells (10,000 cells per well) were seeded into a 96-well plate. Serially diluted virus stocks were inoculated into the cells and incubated at 37 °C for 3 days. The cells were observed under microscopy to judge the CPE appearance. The value of $TCID_{50}$ per ml was calculated using the Reed–Muench method[36].

A B.1.1.7/Alpha isolate (strain QHN001; GISID ID: EPI_ISL_804007) and a B.1.351/Beta isolate (strain TY8-612; GISAID ID: EPI_ISL_1123289) were provided by the National Institute for Infectious Diseases, Japan. The working viruses of these isolates were prepared as described above.

### SARS-CoV-2 infection
One day before infection, Vero cells (10,000 cells), VeroE6/TMPRSS2 cells (10,000 cells) and Calu-3 cells (10,000 cells) were seeded into a 96-well plate. SARS-CoV-2 was inoculated and incubated at 37 °C for 1 h. The infected cells were washed, and 180 µl of culture medium was added. The culture supernatant (10 µl) was collected at the indicated timepoints, and RT–qPCR was used to quantify the viral RNA copy number (see below). To monitor the syncytium formation in infected cell culture, bright-field photos were obtained using ECLIPSE Ts2 (Nikon). The sizes of floating syncytia were measured using the 'quick selection tool' in Photoshop 2020 v.21.0.2 (Adobe) as pixels, and the areas of floating syncytia were calculated from the pixel value. For the GFP-expressing recombinant viruses (Extended Data Fig. 6c), bright-field and green fluorescence images were obtained at the indicated timepoints using an All-in-One Fluorescence microscope BZ-X800 (Keyence), and the GFP fluorescence intensity was analysed using the BZ-X800 Analyzer v.1.1.2.4 (Keyence).

For the infection experiment primary human nasal epithelial cells (Extended Data Fig. 7b), the working viruses were diluted with Opti-MEM (Thermo Fisher Scientific, 11058021). The diluted viruses (1,000 $TCID_{50}$ in 100 µl) were inoculated onto the apical side of the culture and incubated at 37 °C for 1 h. The inoculated viruses were removed and washed twice with Opti-MEM. To collect the viruses on the apical side of the culture, 100 µl Opti-MEM was applied onto the apical side of the culture and incubated at 37 °C for 10 min. Bright-field and green fluorescence images were obtained using ECLIPSE Ts2 (Nikon). The Opti-MEM applied was collected and we used RT–qPCR to quantify the viral RNA copy number (see below).

### Immunofluorescence staining
One day before infection, VeroE6/TMPRSS2 cells (10,000 cells) were seeded into 96-well glass-bottom black plates and infected with SARS-CoV-2 (100 $TCID_{50}$). At 24 h.p.i., the cells were fixed with 4% paraformaldehyde in phosphate-buffered saline (PBS) (Nacalai Tesque, 09154-85) for 1 h at 4 °C. The fixed cells were permeabilized with 0.2% Triton X-100 in PBS for 1 h, blocked with 10% FBS in PBS for 1 h at 4 °C. The fixed cells were then stained using rabbit anti-SARS-CoV-2 N polyclonal antibody (GeneTex, GTX135570) for 1 h. After washing three times with PBS, cells were incubated with an Alexa 488-conjugated anti-rabbit IgG antibody (Thermo Fisher Scientific, A-11008) for 1 h. Nuclei were stained with DAPI (Thermo Fisher Scientific, 62248). Fluorescence microscopy was performed on an All-in-One Fluorescence Microscope BZ-X800 (Keyence).

### Plaque assay
Plaque assay (Extended Data Fig. 2a, c) was performed as previously described[37]. In brief, one day before infection, 200,000 VeroE6/TMPRSS2 cells were seeded into a 12-well plate. The virus was diluted with serum-free Dulbecco's modified Eagle's medium (low glucose) (Wako, 041-29775) containing 1% PS and 20 mM HEPES. After removing the culture medium, the cells were infected with 500 µl of the diluted virus at 37 °C. At 2 h.p.i., 1 ml of mounting solution containing 3% FCS and 1.5% carboxymethyl cellulose (Sigma-Aldrich, C9481-500G) was

overlaid, followed by incubation at 37 °C. At 3 d.p.i., the culture medium was removed, and the cells were washed three times with PBS containing 0.9 mM calcium chloride and 0.5 mM magnesium chloride and fixed with 10% formaldehyde neutral buffer solution (Nacalai Tesque, 37152-51). The fixed cells were washed with tap water, dried and stained with staining solution (2% crystal violet (Nacalai Tesque, 09804-52) in water) for 30 min. The stained cells were washed with tap water and dried, and the size of plaques was measured using ImageJ.

## SARS-CoV-2 reverse genetics

Recombinant SARS-CoV-2 was generated by circular polymerase extension reaction (CPER) as previously described[32,37]. In brief, nine DNA fragments encoding the partial genome of SARS-CoV-2 (strain WK-521, PANGO lineage A; GISAID ID: EPI_ISL_408667)[29] were prepared by PCR using PrimeSTAR GXL DNA polymerase (Takara, R050A). A linker fragment encoding hepatitis delta virus ribozyme, bovine growth hormone polyA signal and cytomegalovirus promoter was also prepared by PCR. A summary of the corresponding SARS-CoV-2 genomic region and the PCR templates and primers used for this procedure is provided in Supplementary Table 3. The ten obtained DNA fragments were mixed and used for CPER[32]. To prepare GFP-expressing replication-competent recombinant SARS-CoV-2, we used fragment 9, in which the *GFP* gene was inserted into the *ORF7a* frame, instead of the authentic F9 fragment[32] (Supplementary Table 3).

To produce recombinant SARS-CoV-2, the CPER products were transfected into HEK293-C34 cells using TransIT-LT1 (Takara, MIR2300) according to the manufacturer's protocol. At 1 day after transfection, the culture medium was replaced with Dulbecco's modified Eagle's medium (high glucose) (Sigma-Aldrich, R8758-500ML) containing 2% FCS, 1% PS and doxycycline (1 μg ml$^{-1}$; Takara, 1311N). At 6 days after transfection, the culture medium was collected and centrifuged, and the supernatants were collected as the seed virus. To remove the CPER products (that is, SARS-CoV-2-related DNA), 1 ml of the seed virus was treated with 2 μl TURBO DNase (Thermo Fisher Scientific, AM2238) and incubated at 37 °C for 1 h. Complete removal of the CPER products (that is, SARS-CoV-2-related DNA) from the seed virus was verified by PCR. The working virus stock was prepared from the seed virus as described above.

To generate recombinant SARS-CoV-2 mutants, mutations were inserted in fragment 8 (Supplementary Table 3) using the GENEART site-directed mutagenesis system (Thermo Fisher Scientific, A13312) according to the manufacturer's protocol with the following primers: Fragment 8_S D614G forward, 5′-CCAGGTTGCTGTTCT TTATCAGGGTGTTAACTGCACAGAAGTCCCTG-3′; Fragment 8_S D614G reverse, 5′-CAGGGACTTCTGTGCAGTTAACACCCTGATAAAGAACA GCAACCTGG-3′; Fragment 8_S P681R forward, 5′-AGACTCAGACT AATTCTCGTCGGCGGGCACGTAGTGTA-3′; and Fragment 8_S P681R reverse, 5′-TACACTACGTGCCCGCCGACGAGAATTAGTCTGAGT CT-3′, according to the manufacturer's protocol. Nucleotide sequences were determined by a DNA sequencing service (Fasmac), and the sequencing data were analysed using Sequencher v.5.1 (Gene Codes). CPER for the preparation of SARS-CoV-2 mutants was performed using mutated fragment 8 instead of parental fragment 8. Subsequent experimental procedures were the same as those for parental SARS-CoV-2 preparation described above. To verify insertion of the mutation into the working viruses, viral RNA was extracted using the QIAamp viral RNA mini kit (Qiagen, 52906) and reverse-transcribed using Super-Script III reverse transcriptase (Thermo Fisher Scientific, 18080085) according to the manufacturers' protocols. DNA fragments including the inserted mutations were obtained by RT–PCR using PrimeSTAR GXL DNA polymerase (Takara, R050A) and the following primers: WK-521 23339-23364 forward, 5′-GGTGGTGTCAGTGTTATAACACCAGG-3′; and WK-521 24089-24114 reverse, 5′-CAAATGAGGTCTCTAGCAGCAATATC-3′. Nucleotide sequences were confirmed as described above, and sequence chromatograms (Extended Data Fig. 2b) were visualized using the web application Tracy (https://www.gear-genomics.com/teal/)[38].

## Viral genome sequencing analysis

The sequences of the working viruses were verified by viral RNA-sequencing analysis. Viral RNA was extracted using the QIAamp viral RNA mini kit (Qiagen, 52906). The sequencing library for total RNA sequencing was prepared using the NEB Next Ultra RNA Library Prep Kit for Illumina (New England Biolabs, E7530). Paired-end, 150 bp sequencing was performed using MiSeq (Illumina) with the MiSeq reagent kit v3 (Illumina, MS-102-3001). Sequencing reads were trimmed using fastp (v.0.21.0)[39] and subsequently mapped to the viral genome sequences of a lineage A isolate (strain WK-521; GISIAD ID: EPI_ISL_408667)[29] or a *GFP*-inserted WK-521 (ref. [32]) using BWA-MEM (v.0.7.17)[40]. Variant calling, filtering and annotation were performed using SAMtools (v.1.9)[41] and snpEff (v.5.0e)[42]. For the clinical isolates (a B.1.617.2/Delta isolate (strain TKYTK1734; GISIAD ID: EPI_ISL_2378732) and a D614G-bearing B.1.1 isolate (strain TKYE610670; GISIAD ID: EPI_ISL_479681)), the detected variants that were present in the original sequences were excluded. Information on the detected mutations is summarized in Supplementary Table 4, and the raw data are deposited at the Gene Expression Omnibus (GSE182738).

## RT–qPCR

RT–qPCR was performed as previously described[37,43]. In brief, 5 μl of culture supernatant was mixed with 5 μl of 2× RNA lysis buffer (2% Triton X-100, 50 mM KCl, 100 mM Tris-HCl (pH 7.4), 40% glycerol, 0.8 U μl$^{-1}$ recombinant RNase inhibitor (Takara, 2313B)] and incubated at room temperature for 10 min. RNase-free water (90 μl) was added, and the diluted sample (2.5 μl) was used as the template for RT–qPCR, which was performed according to the manufacturer's protocol using the One Step TB Green PrimeScript PLUS RT-PCR kit (Takara, RR096A) and the following primers: forward *N*, 5′-AGCCTCTTCTCGTTCCTCATCAC-3′; and Reverse *N*, 5′-CCGCCATTGCCAGCCATTC-3′. The viral RNA copy number was standardized using a SARS-CoV-2 direct detection RT–qPCR kit (Takara, RC300A). Fluorescent signals were acquired using the QuantStudio 3 Real-Time PCR system (Thermo Fisher Scientific), a CFX Connect Real-Time PCR Detection system (Bio-Rad), an Eco Real-Time PCR System (Illumina) or a 7500 Real Time PCR System (Applied Biosystems).

## Plasmid construction

Plasmids expressing the SARS-CoV-2 S proteins of parental D614G (pC-SARS2-S D614G), B.1.1.7/Alpha (pC-SARS2-S Alpha), B.1.351/Beta (pC-SARS2-Beta) and B.1.617.2/Delta (pC-SARS2-S Delta) were prepared in a previous study[31,44]. A plasmid expressing the SARS-CoV-2 S D614G/P681R mutant was generated by site-directed mutagenesis PCR using pC-SARS2-S D614G[31] as the template and the following primers: P681R Fw, 5′-CCAGACCAACAGCCGGAGGAGGGCAAGGTCT-3′ and P681R Rv, 5′-AGACCTTGCCCTCCTCCGGCTGTTGGTCTGG-3′. The resulting PCR fragment was digested with KpnI and NotI and inserted into the KpnI-NotI site of the pCAGGS vector[45].

## Pseudovirus assay

Pseudovirus assays were performed as previously described[31,32,37]. In brief, lentivirus (HIV-1)-based luciferase-expressing reporter viruses pseudotyped with the SARS-CoV-2 S protein and its derivatives, HEK293T cells (1 × 10$^6$ cells), were cotransfected with 1 μg of psPAX2-IN/HiBiT[46], 1 μg of pWPI-Luc2[46] and 500 ng of plasmids expressing parental S or its derivatives using Lipofectamine 3000 (Thermo Fisher Scientific, L3000015) or PEI Max (Polysciences, 24765-1) according to the manufacturer's protocol. At 2 days after transfection, the culture supernatants were collected and centrifuged. The amount of pseudovirus prepared was quantified using the HiBiT assay as previously described[31,46]. The prepared pseudoviruses were stored at −80 °C until use. For the experiment, HOS-ACE2 cells and HOS-ACE2/TMPRSS2 cells (10,000 cells per 50 μl) were seeded in 96-well plates and infected with

100 μl of pseudoviruses prepared at four different doses. At 2 d.p.i., the infected cells were lysed with a One-Glo luciferase assay system (Promega, E6130), and the luminescent signal was measured using a CentroXS3 plate reader (Berthhold Technologies) or GloMax explorer multimode microplate reader 3500 (Promega).

## Western blot analysis

Western blotting was performed as previously described[47–49]. To quantify the level of the cleaved S2 protein in the cells, the collected cells were washed and lysed in lysis buffer (25 mM HEPES (pH 7.2), 20% glycerol, 125 mM NaCl, 1% Nonidet P40 substitute (Nacalai Tesque, 18558-54), protease inhibitor cocktail (Nacalai Tesque, 03969-21)). After quantification of total protein by protein assay dye (Bio-Rad, 5000006), lysates were diluted with 2× sample buffer (100 mM Tris-HCl (pH 6.8), 4% SDS, 12% β-mercaptoethanol, 20% glycerol, 0.05% bromophenol blue) and boiled for 10 min. Then, 10 μl samples (50 μg of total protein) were analysed using western blotting. To quantify the level of the cleaved S2 protein in the virions, 900 μl of the culture medium containing the pseudoviruses was layered onto 500 μl of 20% sucrose in PBS and centrifuged at 20,000g for 2 h at 4 °C. Pelleted virions were resuspended in 1× NuPAGE LDS sample buffer (Thermo Fisher Scientific, NP0007) containing 2% β-mercaptoethanol, and the lysed virions were analysed using western blotting. For protein detection, the following antibodies were used: mouse anti-SARS-CoV-2 S monoclonal antibody (1A9, GeneTex, GTX632604), rabbit anti-ACTB monoclonal antibody (13E5, Cell Signalling, 4970), mouse anti-HIV-1 p24 monoclonal antibody (183-H12-5C, obtained from the HIV Reagent Program, NIH, ARP-3537), horseradish peroxidase (HRP)-conjugated donkey anti-rabbit IgG polyclonal antibody (Jackson ImmunoResearch, 711-035-152) and HRP-conjugated donkey anti-mouse IgG polyclonal antibody (Jackson ImmunoResearch, 715-035-150). Chemiluminescence was detected using SuperSignal West Femto Maximum Sensitivity Substrate (Thermo Fisher Scientific, 34095) or Western BLoT Ultra Sensitive HRP Substrate (Takara, T7104A) according to the manufacturer's instructions. Bands were visualized using the Amersham Imager 600 (GE Healthcare), and the band intensity was quantified using Image Studio Lite v.5.2 (LI-COR Biosciences) or ImageJ v.2.2.0.

## SARS-CoV-2 S-based fusion assay

The SARS-CoV-2 S-based fusion assay was performed as previously described[37]. This assay uses a dual split protein (DSP) encoding *Renilla* luciferase (RL) and *GFP* genes; the respective split proteins, $DSP_{1–7}$ and $DSP_{8–11}$, are expressed in effector and target cells by transfection[48,50]. In brief, on day 1, effector cells (that is, S-expressing cells) and target cells (that is, ACE2-expressing cells) were prepared at a density of $0.6–0.8 × 10^6$ cells in a six-well plate. To prepare effector cells, HEK293 cells were cotransfected with 400 ng of the S expression plasmids and 400 ng $pDSP_{1–7}$ using TransIT-LT1 (Takara, MIR2300). To prepare the target cells, HEK293 cells were cotransfected with pC-ACE2 (0 ng, 200 ng or 1,000 ng) and $pDSP_{8–11}$ (400 ng). Target cells in selected wells were cotransfected with pC-TMPRSS2 (40 ng) in addition to the abovementioned plasmids. On day 3 (24 h after transfection), 16,000 effector cells were detached and reseeded into 96-well black plates (PerkinElmer, 6005225), and target cells were reseeded at a density of 1,000,000 cells per 2 ml per well in six-well plates. On day 4 (48 h after transfection), the target cells were incubated with EnduRen live cell substrate (Promega, E6481) for 3 h and then detached, and 32,000 target cells were added to a 96-well plate with effector cells. RL activity was measured at the indicated timepoints using a Centro XS3 LB960 (Berthhold Technologies). The S proteins expressed on the surfaces of effector cells were stained with rabbit anti-SARS-CoV-2 S monoclonal antibody (HL6, GeneTex, GTX635654) or rabbit anti-SARS-CoV-2 S S1/S2 polyclonal antibody (Thermo Fisher Scientific, PA5-112048). Normal rabbit IgG (Southern-Biotech, 0111-01) was used as a negative control, and APC-conjugated goat anti-rabbit IgG polyclonal antibody (Jackson ImmunoResearch, 111-136-144) was used as a secondary antibody. Expression levels of surface S proteins were analysed using FACS Canto II (BD Biosciences) and FlowJo v.10.7.1 (BD Biosciences). RL activity was normalized to the mean fluorescence intensity of surface S proteins, and the normalized values are shown as fusion activity.

## Mathematical modelling for fusion velocity quantification

The following cubic polynomial regression model was fitted to each of the time-series datasets (Fig. 3e):

$$y ≈ b_0 + b_1x + b_2x^2 + b_3x^3$$

The initial velocity of cell fusion was estimated from the derivative of the fitted cubic curve.

## Neutralization assay

A virus neutralization assay was performed on HOS-ACE2/TMPRSS2 cells using SARS-CoV-2 S pseudoviruses expressing luciferase (see the 'Pseudovirus assay' section). The viral particles that were pseudotyped with D614G S or D614G/P681R S were incubated with serial dilutions of heat-inactivated human serum samples or three receptor-binding-domain-targeting neutralizing antibodies (8A5, Elabscience, E-AB-V1021; 4A3, Elabscience, E-AB-V1024; and CB6, Elabscience, E-AB-V1028) at 37 °C for 1 h. Pseudoviruses without sera and neutralizing antibodies were also included. An 80 μl mixture of pseudovirus and sera/neutralizing antibodies was then added to HOS-ACE2/TMPRSS2 cells (10,000 cells per 50 μl) in a 96-well white plate, and the luminescence was measured as described above (see the 'Pseudovirus assay' section). The 50% neutralization titre ($NT_{50}$) was calculated using Prism 9 software v.9.1.1 (GraphPad Software).

## Statistics and reproducibility

In the time-course experiments using hamsters (Figs. 2c, d, f, 4a, b), two types of statistical tests were performed. First, to evaluate the difference between experimental conditions through all timepoints, a multiple regression analysis including experimental conditions as explanatory variables and timepoints as qualitative control variables was performed. P values were calculated using two-sided Wald tests. In Fig. 2f, family-wise error rates were calculated using the Holm method. Second, to evaluate the difference between two conditions at each timepoint, two-sided Student's t-tests were performed. The data were analysed using Excel v.16.16.8 (Microsoft) or Prism 9 v.9.1.1 (GraphPad Software).

In Fig. 4d and Extended Data Fig. 5, the photographs shown are the representative areas of two independent experiments using 3 hamsters (6 lungs) at each timepoint. In Extended Data Fig. 3, assays were performed in quadruplicate. Photographs shown are the representative of 40 fields of view taken for each sample.

## Reporting summary

Further information on research design is available in the Nature Research Reporting Summary linked to this paper.

## Data availability

The raw data of virus sequences analysed in this study have been deposited at the Gene Expression Omnibus (GSE182738). Publicly available viral sequencing data are available from the GISAID database (https://www.gisaid.org). Source data are provided with this paper.

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

**Acknowledgements** We thank all of the members belonging to The Genotype to Phenotype Japan (G2P-Japan) Consortium; the staff at the National Institute for Infectious Diseases, Japan, and J. Gohda (The University of Tokyo, Japan) for providing virus isolates and reagents; and all of the COVID-19 vaccine study participants at Kyoto University Hospital. An anti-HIV-1 p24 monoclonal antibody (183-H12-5C, ARP-3537) was obtained through the NIH HIV Reagent Program, NIAID, NIH (contributed by B. Chesebro and K. Wehrly). The supercomputing resource was provided by the Human Genome Center at The University of Tokyo and the NIG supercomputer at ROIS National Institute of Genetics. This study was supported in part by AMED Research Program on Emerging and Re-emerging Infectious Diseases (20fk0108163, to A.S.; 20fk0108401, to T.F.; 21fk0108617, to T.F.; 19fk0108113, to Y. Kawaoka; JP20fk0108412, to Y. Kawaoka; 20fk0108146, to K. Sato; 20fk0108270, to K. Sato; 20fk0108413, to T. Ikeda, S.N. and K. Sato; and 20fk0108451, to A.S., T. Irie, A.T.-K., G2P-Japan Consortium, S.N., T. Ikeda, T.F. and K. Sato); the AMED Research Program on HIV/AIDS (21fk0410033, to A.S.; and 21fk0410039, to K. Sato); the AMED Japan Program for Infectious Diseases Research and Infrastructure (20wm0325009, to A.S.; 21wm0325009, to A.S.; and 21wm0125002, to Y. Kawaoka); JST A-STEP (JPMJTM20SL, to T. Ikeda); JST SICORP (e-ASIA) (JPMJSC20U1, to K. Sato); JST SICORP (JPMJSC21U5, to K. Sato), JST CREST (JPMJCR20H6, to S.N.; and JPMJCR20H4, to K. Sato); JSPS KAKENHI Grant-in-Aid for Scientific Research C (19K06382, to A.S.; 18K07156, to K. Tokunaga; and 21K07060, to K. Tokunaga), Scientific Research B (18H02662, to K. Sato; and 21H02737, to K. Sato); the JSPS Fund for the Promotion of Joint International Research (Fostering Joint International Research) (18KK0447, to K. Sato); the JSPS Core-to-Core Program (JPJSCCA20190008, A. Advanced Research Networks, to K. Sato); the JSPS Research Fellow DC1 19J20488 (to I.K.); the JSPS Leading Initiative for Excellent Young Researchers (LEADER) (to T. Ikeda); the ONO Medical Research Foundation (to K. Sato); the Ichiro Kanehara Foundation (to K. Sato); the Lotte Foundation (to K. Sato); the Mochida Memorial Foundation for Medical and Pharmaceutical Research (to K. Sato); the Daiichi Sankyo Foundation of Life Science (to K. Sato); the Sumitomo Foundation (to K. Sato); the Uehara Foundation (to K. Sato); the Takeda Science Foundation (to T. Ikeda and K. Sato); The Tokyo Biochemical Research Foundation (to K. Sato); the Mitsubishi Foundation (to T. Ikeda); the Shin-Nihon Foundation of Advanced Medical Research (to T. Ikeda); the Tsuchiya Foundation (to T. Irie); a Grant for Joint Research Projects of the Research Institute for Microbial Diseases, Osaka University (to A.S.); an intramural grant from Kumamoto University COVID-19 Research Projects (AMABIE) (to T. Ikeda); the Intercontinental Research and Educational Platform Aiming for Eradication of HIV/AIDS (to T. Ikeda); and the Joint Usage/Research Center program of Institute for Frontier Life and Medical Sciences, Kyoto University (to K. Sato).

**Author contributions** A.S., T. Irie, R. Suzuki, H.N., K.U., Y. Kosugi, I.K., E.P.B., Y.L.T., R. Shimizu, K. Shimizu, R.K., T. Ikeda, T.F. and K. Sato performed cell culture experiments. R. Suzuki, T.M., K.I.-H., M. Ito, S.Y., K. Yoshimatsu, M. Imai, T.F. and Y. Kawaoka performed animal experiments. K. Sadamasu, S.O., T.S., K. Tokunaga, I.Y., H.A., M.N. and K. Yoshimura prepared experimental materials. K. Shirakawa, Y. Kazuma, R.N., Y.H. and A.T.-K. collected clinical samples. J.W. and S.N. performed molecular phylogenetic analysis. M. Tsuda, L.W. and S.T. performed histopathological analysis. S.L. performed microCT imaging analysis. J.I. performed mathematical and statistical analysis. A.S., T. Irie, M. Imai, S.T., S.N., T. Ikeda, T.F. Y. Kawaoka and K. Sato designed the experiments and interpreted the results. K. Sato wrote the original manuscript. All of the authors reviewed and proofread the manuscript. The Genotype to Phenotype Japan (G2P-Japan) Consortium contributed to the project administration.

**Competing interests** The authors declare no competing interests.

**Additional information**
**Correspondence and requests for materials** should be addressed to Shinya Tanaka, So Nakagawa, Terumasa Ikeda, Takasuke Fukuhara, Yoshihiro Kawaoka or Kei Sato.

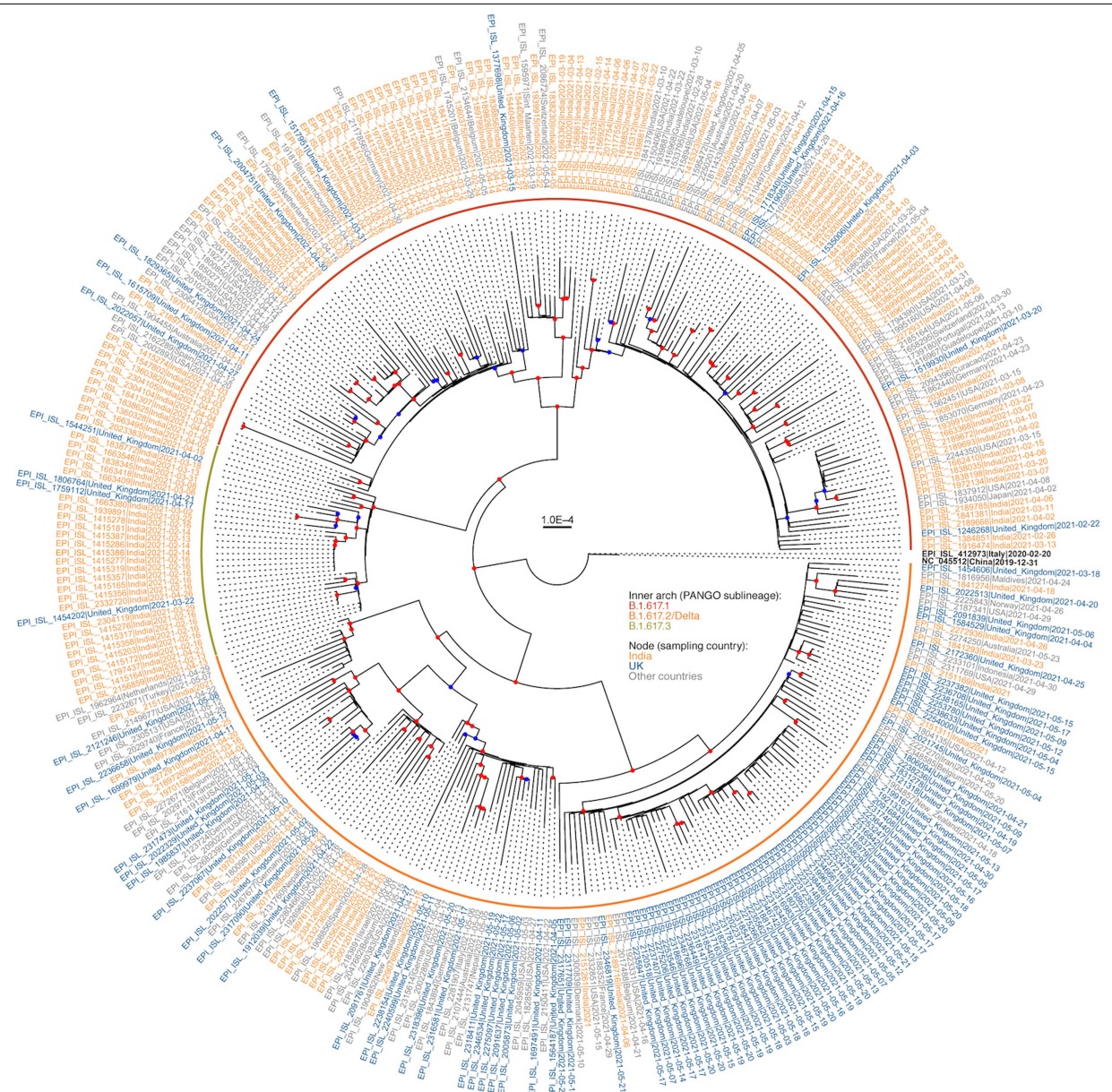

**Extended Data Fig. 1 | A maximum-likelihood-based phylogenetic tree of 334 representative SARS-CoV-2 sequences belonging to the B.1.617 lineage.** GISAID ID, country of exposure, and sampling date were noted in each terminal node. The country isolated (India, the UK, or other countries) and the PANGO sublineage are indicated by the text colour, as indicated in the figure. Coloured circles on the branch are shown on internal nodes for which the bootstrap value was ≥ 80 (red) or ≥ 50 (blue) (n = 1,000).

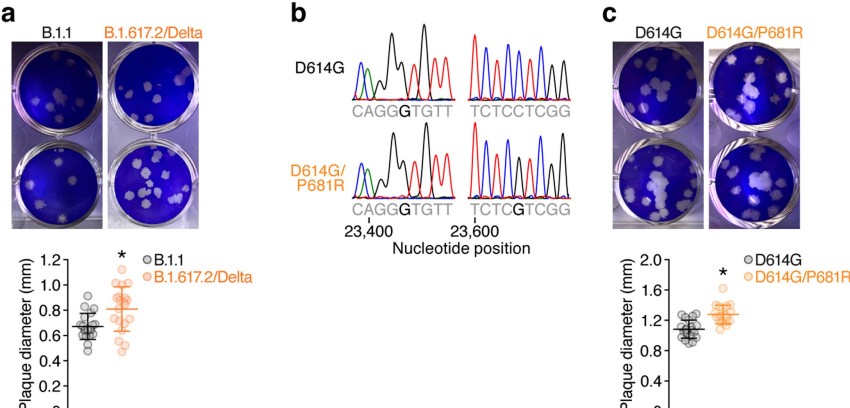

**Extended Data Fig. 2 | Plaques of SARS-CoV-2-infected VeroE6/TMPRSS2 cells. a**, A plaque assay was performed using VeroE6/TMPRSS2 cells as described in **Method**. Representative figures (top) and the summary of the size of plaques (n = 20 for each virus) are shown. Each dot indicates the diameter of the respective plaque. **b**, Chromatograms of nucleotide positions 23,399-23,407 (left) and 23,600-23,608 (right) of parental SARS-CoV-2 (strain WK-521, PANGO lineage A; GISAID ID: EPI_ISL_408667) and the D614G (A23403G in nucleotide)

and P681R (C23604G in nucleotide) mutations. **c**, A plaque assay was performed using VeroE6/TMPRSS2 cells as described in **Method**. Representative figures (top) and the summary of the size of plaques (n = 20 for each virus) are shown. Each dot indicates the diameter of the respective plaque. Data are mean ± S.D (**a**, **c**). Statistically significant differences between B.1.1 and B.1.617.2/Delta (**a**, *P < 0.05) and between D614G and D614G/P681R (**c**, *P < 0.05) were determined by two-sided Mann-Whitney U test.

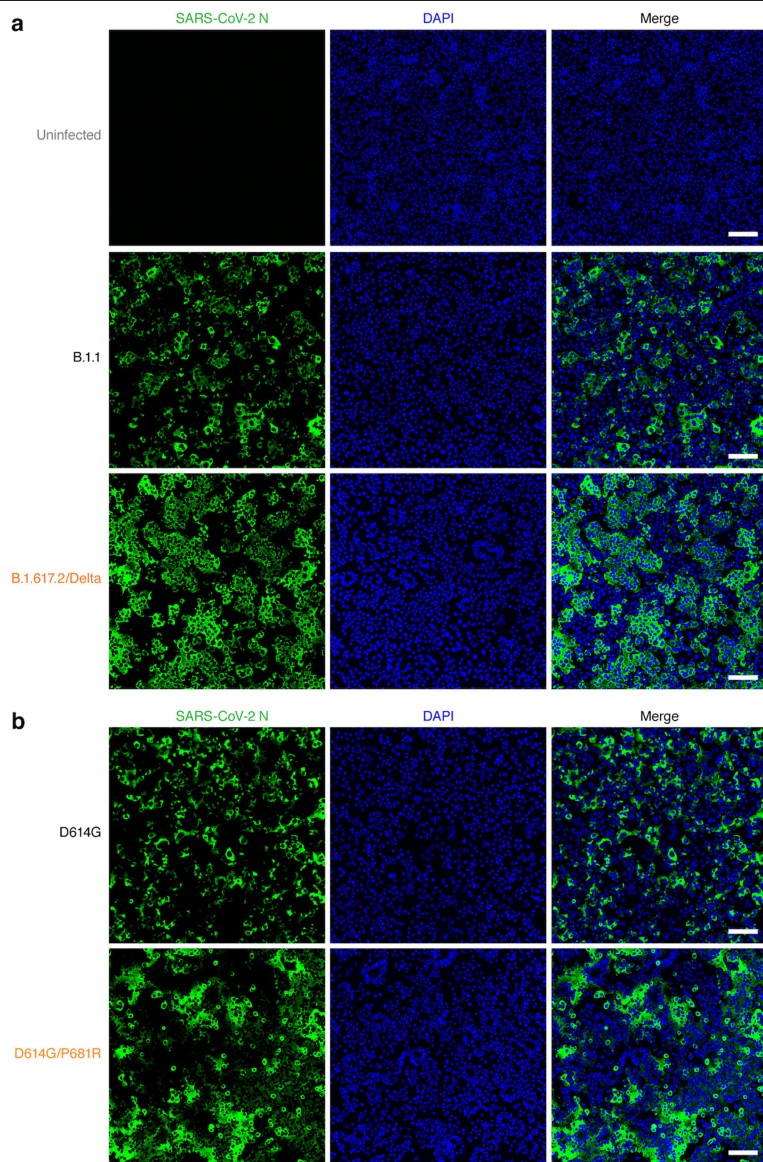

**Extended Data Fig. 3 | Immunofluorescence staining of SARS-CoV-2-infected VeroE6/TMPRSS2 cells.** VeroE6/TMPRSS2 cells were infected with the B.1.1 or B.1.617.2/Delta (**a**) or artificially generated D614G or D614G/P681R (**b**) viruses [multiplicity of infection (MOI) 0.01]. The cells were stained with anti-SARS-CoV-2 nucleocapsid (N) (green) and DAPI (blue). Representative images taken at 24 h.p.i. Bars, 50 μm.

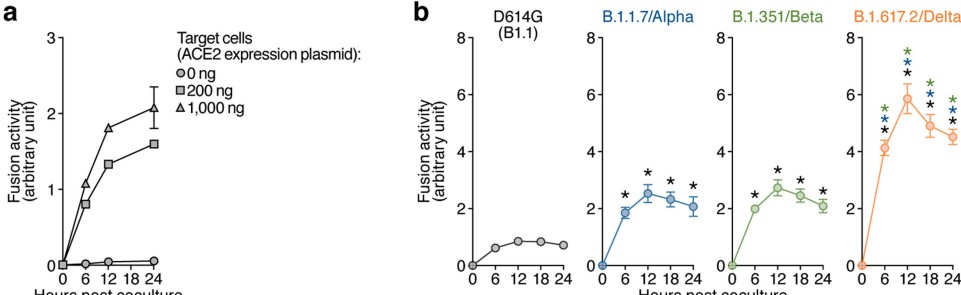

**Extended Data Fig. 4 | SARS-CoV-2 S-based fusion assay. a**, Dependence of human ACE2 expression on the target cells for the SARS-CoV-2 S-based fusion assay. Target cells were prepared by transfecting the indicated amounts of human ACE2 expression plasmid, while Effector cells were prepared by transfecting SARS-CoV-2 S D614G expression plasmid. The fusion activity was measured as described in Methods. Assays were performed in quadruplicate, and fusion activity (arbitrary units) is shown. **b**, Fusogenic activity of the S proteins of VOCs. Effector cells (S-expressing cells) and target cells (ACE2-expressing cells) were prepared, and the fusion activity was measured as described in Methods. Note that the S protein sequence of "D614G" is identical to that of B.1.1 isolate. Assays were performed in quadruplicate, and fusion activity (arbitrary units) is shown. Data are mean ± S.D. In **b**, statistically significant differences (*$P < 0.05$) versus the D614G (black), B.1.1.7/Alpha (blue) or B.1.351/Beta (green) were determined by two-sided Student's $t$ test.

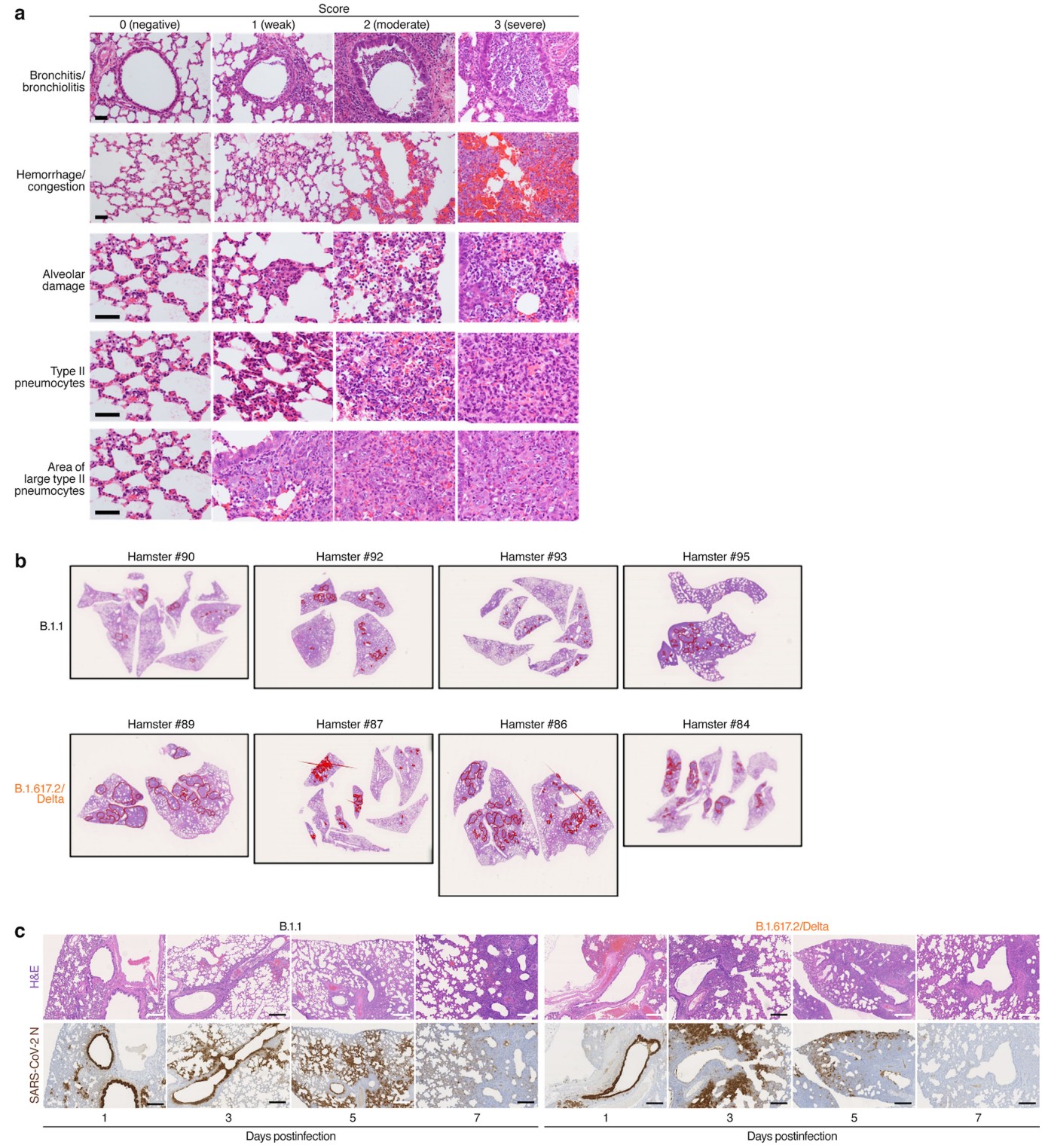

**Extended Data Fig. 5 | Histopathological features of lung lesions.**
**a**, Representative pathological features of lung including bronchitis/
bronchiolitis, haemorrhage/congestion, alveolar damage with apoptosis
and macrophage infiltration, presence of type II pneumocytes, and presence
of the area of large type II pneumocytes are shown. 0 (negative), 1 (weak),
2 (moderate), and 3 (severe). Bars, 50 μm. **b**, Morphometrical analysis of the
area of large type II pneumocytes. The area of the large type II pneumocytes
with the nuclear diameter more than 8 μm in the lung specimens at 5 d.p.i. was
measured, and the percentage of this area in the whole lung tissue area was

calculated. Representative photographs of the lung tissue specimens with B.1.1
isolate (top) and B.1.617.2/Delta isolate (bottom) infections are shown. Red line
indicates the area with the presence of large type II pneumocytes. Note that the
most left panels (hamsters #89 and 90) are identical to the panels shown in
Fig. 2g. **c**, IHC of the viral N proteins in the lung of infected hamsters.
Representative IHC panels of the viral N proteins in the lung of hamsters
infected with D614G-bearing B.1.1 isolate (left) and B.1.617.2/Delta isolates
(right) are shown. Serial sections were used for H&E staining (top) and IHC
(bottom). Bars, 250 μm (1, 3, and 7 d.p.i.) or 500 μm (5 d.p.i.).

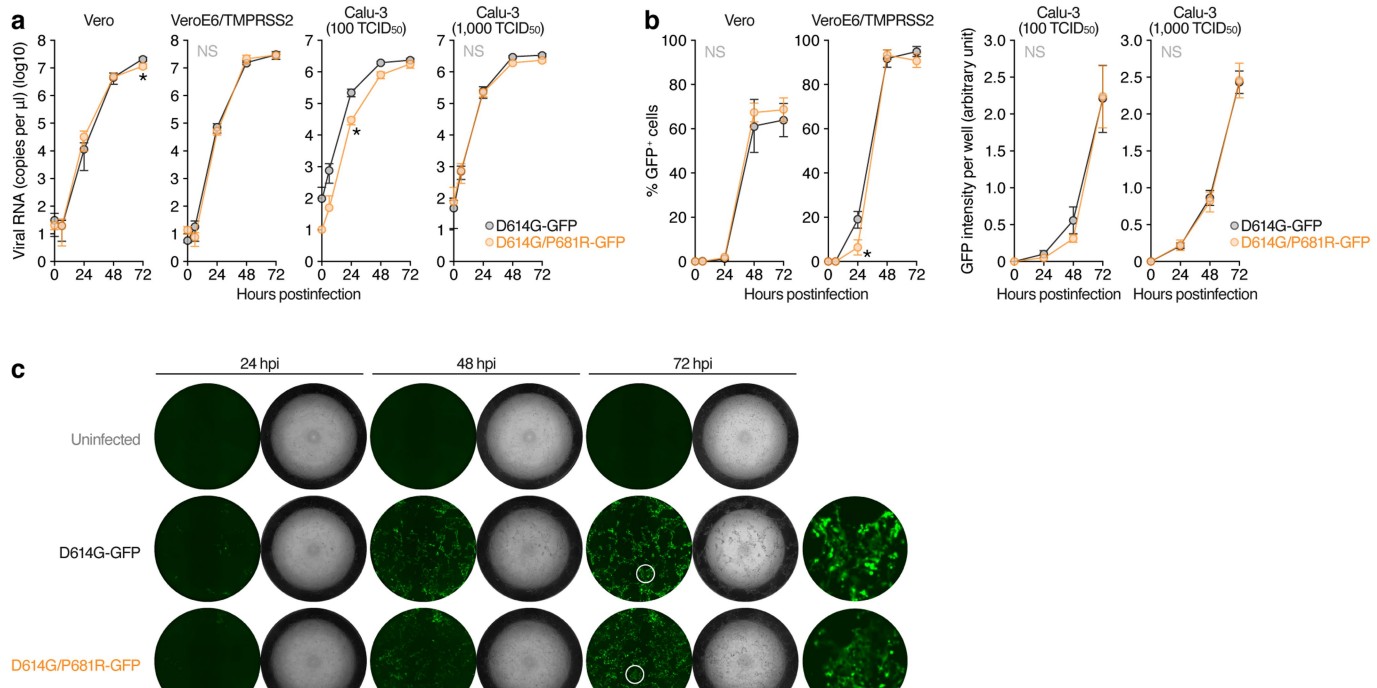

**Extended Data Fig. 6 | Growth kinetics of artificially generated GFP-expressing viruses.** The GFP-expressing D614G and D614G/P681R mutant viruses were generated by reverse genetics. These viruses (100 TCID$_{50}$ for Vero and VeroE6/TMPRSS2 cells, 100 or 1,000 TCID$_{50}$ for Calu-3 cells) were inoculated into cells. The viral RNA copy number of in the culture supernatant (**a**) and the level of GFP-positive cells (the percentage of GFP-positive cells for Vero and VeroE6/TMPRSS2 cells; the GFP intensity per well for Calu-3 cells) (**b**) are shown. Data are mean ± S.D. (**c**) Representative images of Calu-3 cells infected with GFP-expressing viruses (100 TCID$_{50}$). Areas enclosed with circles are enlarged in the right panels. Assays were performed in quadruplicate. Statistically significant differences (*$P < 0.05$) versus the D614G virus were determined by two-sided Student's $t$ test. NS, no statistical significance.

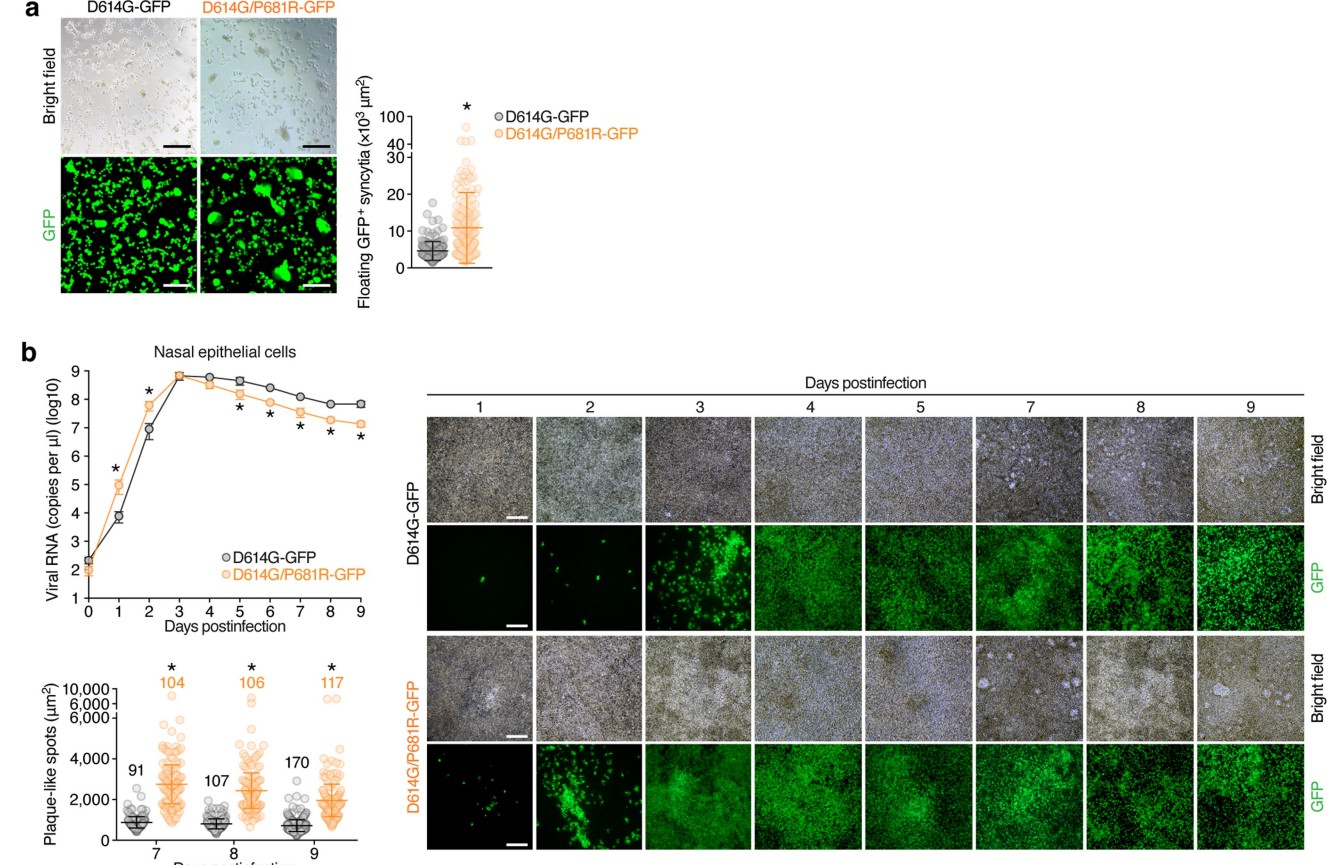

**Extended Data Fig. 7 | Syncytium formation in VeroE6/TMPRSS2 cells infected with GFP-expressing viruses. a**, (Left) Floating syncytia in VeroE6/TMPRSS2 cells infected with GFP-expressing D614G and D614G/P681R mutant viruses (100 $TCID_{50}$) at 72 h.p.i. Bars, 100 µm. (Right) The size distributions of adherent GFP$^+$ syncytia in the D614G mutant-infected (n = 147) and the D614G/P681R mutant-infected (n = 171) cultures. **b**, The GFP-expressing D614G and D614G/P681R mutant viruses (1,000 $TCID_{50}$) were inoculated on the apical side of culture. (Upper left) The copy number of viral RNA on the apical side was quantified as described in Methods, and the growth curves of the inoculated viruses are shown. (Lower left) The size distributions of plaque-like spots in D614G-infected and D614G/P681R-infected cultures. The numbers in the panel indicate the number of plaque-like spots counted. (Right) Time-course of GFP expression. Note that larger plaque-like spots are observed in D614G/P681R-infected culture after 7 d.p.i. Bars, 200 µm. Assays were performed in quadruplicate. Data are mean ± S.D. Statistically significant differences versus D614G (*$P < 0.05$) were determined by two-sided, unpaired Student's $t$ test (**b**, upper left) or the Mann-Whitney U test (**a**, **b**, lower left).

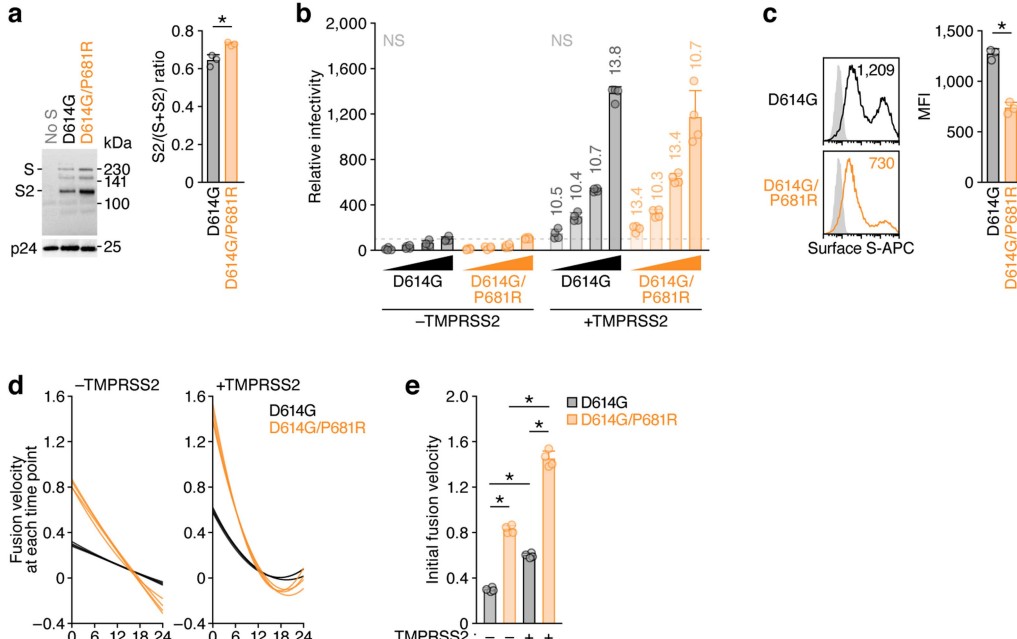

**Extended Data Fig. 8 | Virological phenotypes exhibited by the P681R mutation. a**, Western blotting of pseudoviruses. (Left) Representative blots of SARS-CoV-2 full-length S and cleaved S2 proteins as well as HIV-1 p24 capsid as an internal control. kDa, kilodaltons. (Right) The ratio of S2 to the full-length S plus S2 proteins on pseudovirus particles. Assays were performed in triplicate. Data are mean ± S.D. A statistically significant difference (*, *P* < 0.05) versus D614G S was determined by two-sided Student's *t* test. **b**, Pseudovirus assay. The HIV-1-based reporter virus pseudotyped with SARS-CoV-2 S D614G or D614G/P681R was inoculated into HOS-ACE2 cells or HOS-ACE2/TMPRSS2 cells at 4 different doses (125, 250, 500 and 1,000 ng HIV-1 p24 antigen). Rates of infectivity compared to the virus pseudotyped with parental S D614G (1,000 ng HIV-1 p24) in HOS-ACE2 cells are shown. The labels above the HOS-ACE2/TMPRSS2 bars indicate the fold change versus the corresponding HOS-ACE2. Assays were performed in quadruplicate. **c**, Expression of S protein on the cell surface. (Left) Representative histogram of S protein expression on the cell surface. The number in the histogram indicates the mean fluorescence intensity (MFI). (Right) The MFI of surface S on the S-expressing cells. Assays were performed in triplicate. **d,e**, The kinetics of fusion velocity. **d**, Fitting of a mathematical model based on the kinetics of fusion activity data (see Methods). Each line indicates the result of respective mathematical model on the experimental data (shown in Fig. 3e). **e**, Initial velocity of the S-mediated fusion. Assays were performed in quadruplicate. Data are mean ± S.D. Statistically significant differences (*P* < 0.05) were determined by two-sided, unpaired Student's *t* test without adjustments for multiple comparisons (**b**), two-sided Student's *t* test (**c**) or two-sided Welch's *t* test (**e**). NS, no statistical significance.

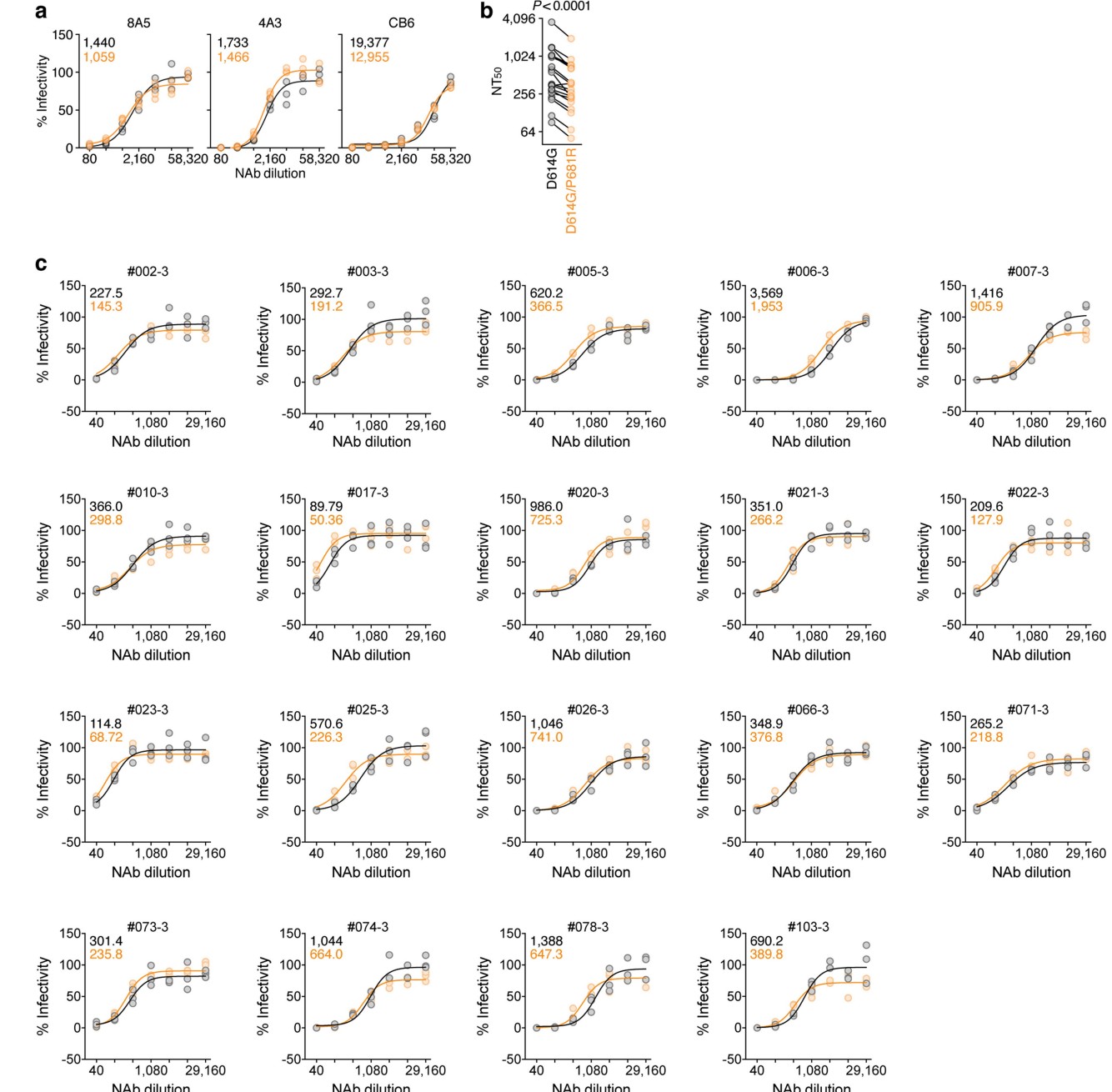

**Extended Data Fig. 9 | Association of the P681R mutation with sensitivity to NAbs. a**, Neutralization assay using three monoclonal antibodies (clones 8A5, 4A3 and CB6). Assays were performed in triplicate. **b**,**c**, Neutralization assay using 19 vaccinated sera. Pseudoviruses and effector cells (S-expressing cells) were treated with serially diluted NAbs or sera as described in Methods. Assays were performed in triplicate. The raw data of panel **b** are shown in panel **c**.

$NT_{50}$, 50% neutralization titre. In **b**, each dot indicates the mean $NT_{50}$ value of the respective donor. A statistically significant difference versus the D614G virus was determined by two-sided Wilcoxon matched-pairs signed rank test. In **c**, the $NT_{50}$ values of D614G S (black) and D614G/P681R S (orange) for each serum are indicated in each panel.

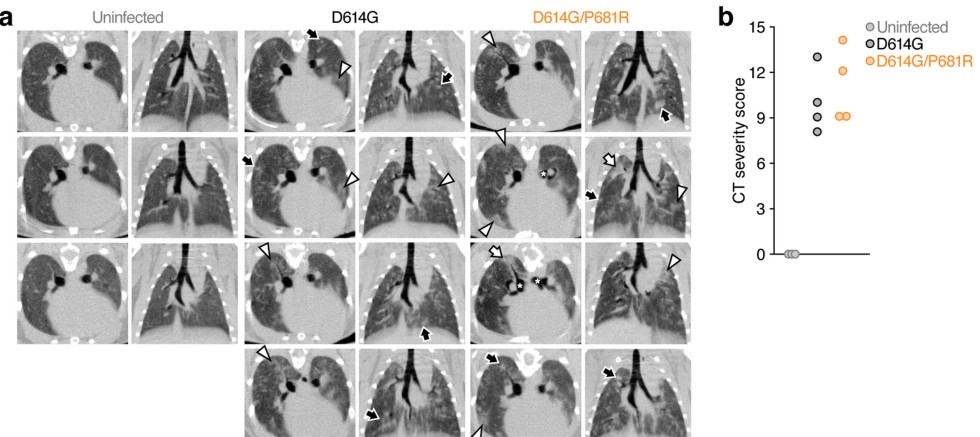

**Extended Data Fig. 10 | MicroCT of the lung of infected hamsters. a**, MicroCT axial and coronal images of the lungs of Syrian hamsters at 7 d.p.i. with D614G-infected (n = 4), D614G/P681R-infected (n = 4), and uninfected hamsters (n = 3). Lung abnormalities included multifocal nodules (black arrows), ground glass opacity (white arrowheads), and regions of lung consolidation (white arrows) that were peripheral, bilateral, and multilobar. Pneumomediastinum is labelled with white asterisks. **b**, Summary of CT severity score. CT severity score of D614G-infected (n = 4), D614G/P681R-infected (n = 4), and uninfected hamsters (n = 3) Syrian hamsters. Each dot indicates the result of the respective infected hamster. Note that D614G/P681R-infected animals had a higher average CT severity score compared to D614G-infected animals.

# Reporting Summary

## Statistics

For all statistical analyses, confirm that the following items are present in the figure legend, table legend, main text, or Methods section.

| n/a | Confirmed | |
|---|---|---|
| ☐ | ☒ | The exact sample size (*n*) for each experimental group/condition, given as a discrete number and unit of measurement |
| ☐ | ☒ | A statement on whether measurements were taken from distinct samples or whether the same sample was measured repeatedly |
| ☐ | ☒ | The statistical test(s) used AND whether they are one- or two-sided <br> *Only common tests should be described solely by name; describe more complex techniques in the Methods section.* |
| ☒ | ☐ | A description of all covariates tested |
| ☒ | ☐ | A description of any assumptions or corrections, such as tests of normality and adjustment for multiple comparisons |
| ☐ | ☒ | A full description of the statistical parameters including central tendency (e.g. means) or other basic estimates (e.g. regression coefficient) AND variation (e.g. standard deviation) or associated estimates of uncertainty (e.g. confidence intervals) |
| ☐ | ☒ | For null hypothesis testing, the test statistic (e.g. *F*, *t*, *r*) with confidence intervals, effect sizes, degrees of freedom and *P* value noted <br> *Give P values as exact values whenever suitable.* |
| ☒ | ☐ | For Bayesian analysis, information on the choice of priors and Markov chain Monte Carlo settings |
| ☒ | ☐ | For hierarchical and complex designs, identification of the appropriate level for tests and full reporting of outcomes |
| ☒ | ☐ | Estimates of effect sizes (e.g. Cohen's *d*, Pearson's *r*), indicating how they were calculated |

*Our web collection on statistics for biologists contains articles on many of the points above.*

## Software and code

Policy information about availability of computer code

| Data collection | MAFFT suite v7.407, IQ-TREE v2.1.3, FinePointe software v2.8.0.12146 (Data Sciences International), CosmoScan Database software v3.3.27.100 (Rigaku Corporation) |
|---|---|
| Data analysis | BZ-X800 Analyzer v1.1.2.4 (Keyence), Image Studio Lite v5.2 (LI-COR Biosciences), Fiji v2.2.0 in ImageJ v2.2.0, CosmoScan Database software v3.3.27.100 (Rigaku Corporation), Sequencher software v5.1 (Gene Codes Corporation), Excel software v16.16.8 (Microsoft), Photoshop 2020 v21.0.2 (Adobe) |

For manuscripts utilizing custom algorithms or software that are central to the research but not yet described in published literature, software must be made available to editors and reviewers. We strongly encourage code deposition in a community repository (e.g. GitHub). See the Nature Portfolio guidelines for submitting code & software for further information.

## Data

Policy information about availability of data

All manuscripts must include a data availability statement. This statement should provide the following information, where applicable:

- Accession codes, unique identifiers, or web links for publicly available datasets
- A description of any restrictions on data availability
- For clinical datasets or third party data, please ensure that the statement adheres to our policy

The raw data of virus sequences analysed in this study are deposited in Gene Expression Omnibus (accession number: GSE182738). Publicly available viral sequence data are available from GISAID database. The accession numbers of viral sequences used in this study are listed in Method section.

# Field-specific reporting

Please select the one below that is the best fit for your research. If you are not sure, read the appropriate sections before making your selection.

☒ Life sciences  ☐ Behavioural & social sciences  ☐ Ecological, evolutionary & environmental sciences

For a reference copy of the document with all sections, see nature.com/documents/nr-reporting-summary-flat.pdf

# Life sciences study design

All studies must disclose on these points even when the disclosure is negative.

| | |
|---|---|
| Sample size | The sample sizes (n > 3) for cell culture experiments were chosen for applying statistical tests. The sample sizes (n > 4) for the hamster studies were chosen because they have previously been shown to be sufficient to evaluate a significant difference among groups (Belser et al., Nature, 2013; Zhang et al., Science, 2013; Imai et al., Nature Microbiology, 2020). |
| Data exclusions | No data were excluded. |
| Replication | In vitro experiments representative of at least 2 experiments with multiple samples per time point. In vivo experiments (hamster) utilized multiple animals per group per time point and were from more than single experiment. In vivo experiments were replicated and performed independently. All attempts at replication were successful. |
| Randomization | No method of randomization was used to determine how the animals were allocated to the experimental groups and processed in this study, because covariates (sex and age) were identical (male, 4 weeks old). For experiments other than animal studies, randomization is not applicable because homogenous materials (i.e., cell lines) were used. Primary human nasal epithelial cells were used in an experiment, but only one donor was used. Therefore, randomization is not applicable. |
| Blinding | For the microCT imaging (Extended Data Fig. 19), images were anonymized and randomized; the scorer was blinded to the group allocation. No blinding was carried out for the other experiments, because these are not relevant for an observational study. |

# Reporting for specific materials, systems and methods

We require information from authors about some types of materials, experimental systems and methods used in many studies. Here, indicate whether each material, system or method listed is relevant to your study. If you are not sure if a list item applies to your research, read the appropriate section before selecting a response.

## Materials & experimental systems

| n/a | Involved in the study |
|---|---|
| ☐ | ☒ Antibodies |
| ☐ | ☒ Eukaryotic cell lines |
| ☒ | ☐ Palaeontology and archaeology |
| ☐ | ☒ Animals and other organisms |
| ☐ | ☒ Human research participants |
| ☒ | ☐ Clinical data |
| ☒ | ☐ Dual use research of concern |

## Methods

| n/a | Involved in the study |
|---|---|
| ☒ | ☐ ChIP-seq |
| ☐ | ☒ Flow cytometry |
| ☒ | ☐ MRI-based neuroimaging |

## Antibodies

| | |
|---|---|
| Antibodies used | For Western blotting:<br>Mouse anti-SARS-CoV-2 S monoclonal antibody (clone 1A9, GeneTex, Cat# GTX632604, 1:20,000)<br>Rabbit anti-ACTB monoclonal antibody (clone 13E5, Cell Signalling, Cat# 4970, 1:5,000)<br>Mouse anti-HIV-1 p24 monoclonal antibody (clone 183-H12-5C, obtained from the HIV Reagent Program, NIH, Cat# ARP-3537, 1:5,000)<br>Horseradish peroxidase (HRP)-conjugated donkey anti-rabbit IgG polyclonal antibody (Jackson ImmunoResearch, Cat# 711-035-152, 1:10,000)<br>HRP-conjugated donkey anti-mouse IgG polyclonal antibody (Jackson ImmunoResearch, Cat# 715-035-150, 1:10,000).<br><br>For flow cytometry:<br>Rabbit anti-SARS-CoV-2 S monoclonal antibody (clone HL6, GeneTex, Cat# GTX635654, 1:200)<br>Rabbit anti-SARS-CoV-2 S S1/S2 polyclonal antibody (Thermo Fisher Scientific, Cat# PA5-112048, 1:100)<br>APC-conjugated goat anti-rabbit IgG polyclonal antibody (Jackson ImmunoResearch, Cat# 111-136-144, 1:100)<br><br>Normal rabbit IgG (SouthernBiotech, Cat# 0111-01, 1:100) |

For immunohistolochemistry:
Mouse anti-SARS-CoV-2 N monoclonal antibody (clone 1035111, R&D systems, Cat# MAB10474-SP, 1:400)

For neutralisation assay:
Human anti-SARS-CoV-2 RBD neutralizing antibodies (clone 8A5, Elabscience, Cat# E-AB-V1021; clone 4A3, Elabscience, Cat# E-AB-V1024; and clone CB6, Elabscience, Cat# E-AB-V1028)

| Validation | Validation was conducted by manufacturers prior to sale, and validation statements are available on the manufacturers' website. |
|---|---|

# Eukaryotic cell lines

Policy information about cell lines

| Cell line source(s) | HEK293 cells (a human embryonic kidney cell line; ATCC CRL-1573)<br>HEK293T cells (a human embryonic kidney cell line; ATCC CRL-3216)<br>HOS cells (a human osteosarcoma cell line; ATCC CRL-1543)<br>Vero cells [an African green monkey (Chlorocebus sabaeus) kidney cell line; JCRB0111]<br>VeroE6/TMPRSS2 cells [an African green monkey (Chlorocebus sabaeus) kidney cell line; JCRB1819]<br>Calu-3 cells (a human lung epithelial cell line; ATCC HTB-55)<br>HEK293-C34 cells (Torii et al., Cell Reports, 2020)<br>HOS-ACE2/TMPRSS2 cells: prepared as previously described (Ferreira et al., bioRxiv, 2021; Ozono et al., Nat Commun, 2021).<br>Primary human nasal epithelial cells: purchased from Epithelix (Cat# EP01, Batch# MD0436). |
|---|---|
| Authentication | None of the cells used were authenticated. |
| Mycoplasma contamination | All cell lines were regularly tested for mycoplasma contamination by using PCR and were confirmed to be mycoplasma-free. |
| Commonly misidentified lines<br>(See ICLAC register) | No commonly misidentified cell lines were used. |

# Animals and other organisms

Policy information about studies involving animals; ARRIVE guidelines recommended for reporting animal research

| Laboratory animals | Syrian hamsters (male, 4 weeks old) were purchased from Japan SLC Inc. (Shizuoka, Japan). |
|---|---|
| Wild animals | No wild animal was used in this study. |
| Field-collected samples | No field collected sample was used in the study. |
| Ethics oversight | All experiments with hamsters were performed in accordance with the Science Council of Japan's Guidelines for Proper Conduct of Animal Experiments. The protocols were approved by the Institutional Animal Care and Use Committee of National University Corporation Hokkaido University (approval number 20-0123) and the Animal Experiment Committee of the Institute of Medical Science, the University of Tokyo (approval number PA19-75). |

Note that full information on the approval of the study protocol must also be provided in the manuscript.

# Human research participants

Policy information about studies involving human research participants

| Population characteristics | Peripheral blood was collected four weeks after the second vaccination with BNT162b2 (Pfizer-BioNTech), and sera were isolated from the peripheral blood of 19 vaccinees (average age: 38, range: 28-59, 26% male). |
|---|---|
| Recruitment | The voluntary donors were the Kyoto University Hospital staffs four weeks after the second vaccination with BNT162b2. The donors were recruited by massive mail invitation regardless of age, sex, gender, race, ethnicity, or other characteristics. Written informed consent was obtained from the voluntary donor. The sera used in this study were selected randomly. |
| Ethics oversight | All protocols involving specimens from human subjects recruited at Kyoto University were reviewed and approved by the Institutional Review Boards of Kyoto University (approval number G0697) and the Institute of Medical Science, the University of Tokyo (approval number 2021-1-0416). |

Note that full information on the approval of the study protocol must also be provided in the manuscript.

# Flow Cytometry

## Plots

Confirm that:

[✗] The axis labels state the marker and fluorochrome used (e.g. CD4-FITC).

[✗] The axis scales are clearly visible. Include numbers along axes only for bottom left plot of group (a 'group' is an analysis of identical markers).

[✗] All plots are contour plots with outliers or pseudocolor plots.

[✗] A numerical value for number of cells or percentage (with statistics) is provided.

## Methodology

| | |
|---|---|
| Sample preparation | HEK293 cells were cotransfected with 400 ng of D614G S or D614G/P681R expression plasmids and 400 ng pDSP1-7 using TransIT-LT1 (Takara, Cat# MIR2300). |
| Instrument | FACS Canto II instrument (BD Biosciences) |
| Software | FlowJo |
| Cell population abundance | 10,000 cells gated in the FSC-A/SSC-A plot (Extended Data Fig. 15a) were acquired for each condition. |
| Gating strategy | Shown in Supplementary Fig. 1. |

[✗] Tick this box to confirm that a figure exemplifying the gating strategy is provided in the Supplementary Information.

