## [Peer Review File · Nature]

Manuscript Title: Enhanced fusogenicity and pathogenicity of SARS-CoV-2 Delta P681R mutation

Reviewer Comments & Author Rebuttals

Reviewer Reports on the Initial Version:

Referee #1 (Remarks to the Author):

This is a timely and important study on the phenotypic assessment of the SARS-CoV-2 Delta variant. The authors present a phylogenetic analysis of the B.1.617 lineage and emphasise the P681R substitution in the spike protein as a hallmark of the B.1.617 lineage. This analysis is followed by a phenotypic analysis of a Delta variant isolate that reveals that this isolate has increased ability to induce syncytia in infected cell layers and that infection in Syrian hamsters shows higher pathogenicity. These features are then assessed in a recombinant P681R mutant virus. Also this mutant virus showed enhanced fusogenicity and pathogenicity.

Overall, the experiments on spike-mediated fusion are well conducted and the evidence that the P681R substitution is a major determinant of the higher fusion activity of the Delta variant spike protein is convincing. However, the major concern and limitation of this study is the assessment of pathogenicity (see specific comment below). This part is less well elaborated and not fully convincing.

Major concern:

1. There's one major concern with the claim that the Delta variant has increased pathogenicity and that the P681R substitution in the spike protein is the determinant of this increased pathogenicity. Although the authors have performed infection studies in hamsters, it seems that only selected data are shown. The claim of increased pathogenicity is mainly based on slightly increased weight loss of Delta or the P681R mutant compared to B.1.1 wt or D614G wt, respectively. Viral RNA loads, virus titers of lungs and nasal turbinate are shown for Delta and P681R respectively, but no complete picture is presented. There're also no pathological studies shown. Based on the provided data the claim of increased pathogenicity is only weakly supported. Additional evidence is needed, and the complete data set should be presented even if differences may be very small.

minor concerns:

1. The authors claim that syncytium formation is increased for Delta and the P681R mutant. Both were compared to well selected control viruses. Nevertheless it would be interesting to directly compare the Delta isolate with the recombinant P681R mutant with respect to fusogenicity.

2. The authors performed a number of experiments illustrating the increased fusogenicity. It would be interesting to add a comment or data if plaque morphology or size is also different. Further, since the authors like to imply that there's an association of the increased fusogenicity and the increased pathogenicity, it would be interesting to assess fusogenicity in a more relevant primary cell culture model, such as primary lung or nasal epithelial cultures.

3. It is not stated anywhere if the virus stocks have been fully sequenced. It is now well known that in particular the FCS region is quite unstable in cell culture. It is therefore important to demonstrate that the viruses used in this study have the expected genome sequence and that there're no, or only minor variants in the stocks.

4. The virus dose used for infection in the hamster experiments vary and is 10^5 TCID₅₀ in the first experiment and 10^4 TCID₅₀ in the second experiment. Is there any reason for selecting two

different dosages?

Referee #2 (Remarks to the Author):

Strength

1. Though the phenomenon of forming larger and more numerous syncytia by B.1.1.7 alpha and B.1.351 beta variants has already been reported, this report is the first one to report the same phenomenon for B.1.617.2/Delta variant. See preprint in <https://www.biorxiv.org/content/10.1101/2021.06.11.448011v1>
2. More importantly, the authors have shown that the P681R caused more weight loss and higher lung viral load in the hamster model.
3. This study on the P681R mutation being the virulence factor of the B.1.617.2/Delta variant is timely and important.

Revisions that can address the following weaknesses before publication:

1. The authors suggest that delta and P681R recombinant virus have an increased pathogenicity in hamsters. This claim needs more support in terms of explaining:
 - a. How did the increased fusion and increased spike cleavage explain the decreased virus replication of delta (as shown in Fig 2c)?
 - b. The only support for the increased pathogenicity comes from Fig 2d, which the authors showed a statistically significant difference between B.1.1 and delta on day 3 and day 4. As explained in the figure legend, there were 12 hamsters in the delta group. However, there is no variation between these 12 hamsters between day 1 to day 4. This is highly unusual. The same problem applies to fig 4a. This would need further explanation and elaboration.
 - c. Based on the modest differences presented in fig2d and fig4a-b, the authors' conclusion on the increased pathogenicity of delta or P681 virus in hamsters needs to be tuned down.
2. The authors suggest that the cell-cell fusion mediated by P681R results in an increased resistance to neutralizing antibodies, which then leads to an increased pathogenicity. This claim may need more evidence to support:
 - a. There is no difference between D614G and D614G/P681R in the fusion experiment in the presence of neutralizing antibodies. The authors need to explain this lack of difference.
 - b. Thus the authors need further experiment to support this claim, which is the key message of this study.
3. At least in the syncytial assays, the alpha and beta variant should be included as controls.

Others:

1. Extended Data Fig 2. At 72 hpi, there are one and three positive cells for B.1.1 and delta virus infected VeroE6/TMPRSS2 in the entire field, respectively. The number of infected cell is disproportionately low even when the cells are infected at 0.01 MOI.
2. Extended Data Fig 8C. All three antibodies increased virus fusion by 100-200% at all dilutions. This is suggesting the presence of a potential systemic problem with the fusion assays that may infringe on the validity of these assays or the claims by the authors.
3. The correct statistical tests should be used instead of Student's t-tests. Two-way ANOVA should be used in Fig 2a, c, d, as well as for Fig 4a-c.

Referee #3 (Remarks to the Author):

Saito et al. report spike mutation P681R enhances the fusogenicity and pathogenicity of Delta SARS-CoV-2. They used cell culture (Vero, Vero expressing TMPRSS, and Calu-3 cells), pseudovirus expressing SARS-CoV-2 spikes, cell-to-cell- fusion assay, and hamster models to analyze the P681R mutation. Overall, the results are interesting and supporting the conclusions. The following points should be addressed to substantiate the study.

Major comments

1. The cell-to-cell fusion experiment should include a control of no-hACE2 expression in target cells. This will examine if hACE2 is required for the cell-to-cell fusion.
2. A more detailed virus entry experiment should be added to clearly demonstrate the potential role of P618R mutation in virus entry (or not). For example, the authors can compare the intracellular viral RNA levels between D614G and D614G/P681R viruses at 0.5 or 1 h post infection.
3. The authors should mention why hamster transmission experiments were not performed or presented.
4. The authors should discuss how these results are associated with the increased transmissibility of Delta variant observed in humans.

Author Rebuttals to Initial Comments:

Referee #1 (Remarks to the Author):

This is a timely and important study on the phenotypic assessment of the SARS-CoV-2 Delta variant. The authors present a phylogenetic analysis of the B.1.617 lineage and emphasise the P681R substitution in the spike protein as a hallmark of the B.1.617 lineage. This analysis is followed by a phenotypic analysis of a Delta variant isolate that reveals that this isolate has increased ability to induce syncytia in infected cell layers and that infection in Syrian hamsters shows higher pathogenicity. These features are then assessed in a recombinant P681R mutant virus. Also this mutant virus showed enhanced fusogenicity and pathogenicity.

Overall, the experiments on spike-mediated fusion are well conducted and the evidence that the P681R substitution is a major determinant of the higher fusion activity of the Delta variant spike protein is convincing. However, the major concern and limitation of this study is the assessment of pathogenicity (see specific comment below). This part is less well elaborated and not fully convincing.

Major concern:

1. There's one major concern with the claim that the Delta variant has increased pathogenicity and that the P681R substitution in the spike protein is the determinant of this increased pathogenicity. Although the authors have performed infection studies in hamsters, it seems that only selected data are shown. The claim of increased pathogenicity is mainly based on slightly increased weight loss of Delta or the P681R mutant compared to B.1.1 wt or D614G wt, respectively. viral RNA loads, virus titers of lungs and nasal turbinates are shown for Delta and P681R respectively, but no complete picture is presented. There're also no pathological studies shown. Based on the provided data the claim of increased pathogenicity is only weakly supported. Additional evidence is needed, and the complete data set should be presented even if differences may be very small.

Our reply:

In the revised manuscript, we additionally presented the pathological data for the lung of the hamsters infected with B.1.1 and B.1.617.2/Delta (**Fig. 2e-2h**). We additionally analyzed the time course of the pathological features by scoring the degree of lung lesions (**Fig. 2f, 2g** of the revised manuscript; **pages 6-7, lines 192-201**). At 5 dpi, enlarged type II pneumocytes were significantly increased in the case of B.1.617.2/Delta infection (**Fig. 2f, 2g** of the revised manuscript). The appearance of type II pneumocytes is one of the defense mechanisms of lung tissue from the damage. Therefore, our additional data should reflect the higher degree of the epithelial damage/pathogenicity by the B.1.617.2/Delta infection compared to B.1.1 virus.

On the other hand, the differences between the D614G/P681R virus and D614G virus in infected hamsters were relatively small compared to those between the B.1.617.2/Delta variant and the B.1.1 virus (**Fig. 4d, 4e** of the revised manuscript). As yet, the increased number of type II pneumocytes in the lung of D614G/P681R-infected hamster at 7 dpi (**Fig. 4d** of the revised manuscript) may reflect the higher degree of the inflammatory reaction than the D614G-infected hamsters. Therefore, in the revised manuscript, the explanation for the findings of the P681R-bearing virus was toned down, according to the suggestion from the referee 2 (see below) (e.g., page 4, lines 111; page 11, lines 364-369; page 12, line 405).

Nevertheless, in the revised manuscript, we additionally performed statistical analyses and showed that both the B.1.617.2/Delta and P681R-bearing viruses are significantly more pathogenic than the parental viruses (**Fig. 2c, 2d, 2f, 4a, 4b**). Therefore, we believe that the increased pathogenicity of the B.1.617.2/Delta variant *in vivo* is clearly shown in our revised manuscript and the P681R mutation is (partly) associated with the higher pathogenicity of this VOC.

minor concerns:

1. The authors claim that syncytium formation is increased for Delta and the P681R mutant. Both were compared to well selected control viruses. Nevertheless it would be interesting to directly compare the Delta isolate with the recombinant P681R mutant with respect to fusogenicity.

Our reply:

Fig. R1. Fusogenic activity of the S proteins of B.1.617.2/Delta and D614G/P681R.

SARS-CoV-2 S-based fusion assay. Effector cells (S-expressing cells) and target cells (ACE2-expressing cells) were prepared, and the fusion activity was measured as described in **Methods**. Note that the S protein sequence of “D614G” is identical to that of B.1.1 isolate. Assays were performed in quadruplicate, and fusion activity (arbitrary units) is shown. Data are mean ± S.D. Statistically significant differences (*, $P < 0.05$) versus the D614G determined by Student's t test.

To address the referee's concern, we performed a cell-based fusion assay using the S proteins of D614G (note that this is identical to the S protein of B.1.1 isolate), B.1.617.2/Delta and D614G/P681R. As shown above, although the fusogenicity of the S proteins of

B.1617.2/Delta and D614G/P681R was significantly higher than that of parental D614G S, these values of B.1617.2/Delta and D614G/P681R were comparable (**Fig. R1**). The comparable fusogenicity of B.1617.2/Delta and D614G/P681R is supported by our findings shown in **Fig. 2b, 3b, 3c, Extended Figs. 2, 3** of the revised manuscript, and the direct comparison between B.1617.2/Delta and D614G/P681R may not be needed in the flowline of current manuscript. Therefore, we did not show **Fig. R1** in the revised manuscript.

Additionally, we repeated the experiments showing syncytia formation, and the data were replaced with the new ones (**Fig. 2b and Fig. 3b** of revised manuscript). As shown below (**Fig. R2**; this is a combined figure of **Fig. 2b and Fig. 3b** of the revised manuscript), the sizes of syncytia formed by the infections of B.1.617.2/Delta and D614G/P681R viruses are comparable, suggesting that the fusogenicity of the Delta variant and the P681R-bearing virus is comparable.

Fig. R2. Syncytium formation.

Syncytia in infected VeroE6/TMPRSS2 cells were observed at 72 hpi. The size distributions of floating syncytia in the cultures infected with B.1.1 (n = 215), B.1.617.2/Delta (n = 216), D614G (n = 228) and D614G/P681R (n = 164) viruses. The size distribution of the floating single cells in uninfected culture (n = 177) is also shown as a negative control. Statistically significant differences versus parental viruses (*, $P < 0.05$) or uninfected cells (#, $P < 0.05$) were determined by the Mann-Whitney U test.

2. The authors performed a number of experiments illustrating the increased fusogenicity. It would be interesting to add a comment or data if plaque morphology or size is also different.

Our reply:

We showed the data that the plaque sizes of the Delta and the P681R-bearing viruses are significantly bigger than their parental viruses (**Extended Data Figs. 2, 9** of the revised manuscript).

Further, since the authors like to imply that there's an association of the increased fusogenicity and the increased pathogenicity, it would be interesting to assess fusogenicity in a more relevant primary cell culture model, such as primary lung or nasal epithelial cultures.

Our reply:

Thank you very much for the important suggestion. In the revised manuscript, we showed

the data of virus replication assay using primary human nasal epithelial cells. As shown in **Fig. 3d** of the revised manuscript, the viral growth during the acute phase of the P681R-bearing virus infection is significantly higher than that of the parental virus infection. We also found that the P681R-bearing virus exhibits plaque-like spots, which would be due to its higher fusogenicity, during late phase of infection (**Fig. 3d** of the revised manuscript).

Regarding the fusion assay using primary cell culture model – the culture method of primary nasal epithelial cells, which is called an "air-liquid interface" culture, is quite different from that of typical lung cell lines. Therefore, it would be technically impossible to perform our fusion assay using the primary culture.

3. it is not stated anywhere if the virus stock have been fully sequenced. It is now well known that in particular the FCS region is quite unstable in cell culture. It is therefore important to demonstrate that the viruses used in this study have the expected genome sequence and that there're no, or only minor variants in the stocks.

Our reply:

Thank you very much for the important suggestion. We analyzed the viral genome sequences by NGS and verified that there are no irrelevant mutations in the S protein-coding region (including the FCS region) of the working viruses used this study. The summarized data were shown in **Extended Data Table 4** of the revised manuscript, and the raw data were deposited in the NCBI database (GEO: GSE182738).

4. The virus dose used for infection in the hamster experiments vary and is 10^5 TCID₅₀ in the first experiment and 10^4 TCID₅₀ in the second experiment. Is there any reason for selecting two different dosages?

Our reply:

This is because these two animal experiments (shown in **Fig. 2 and Fig. 4**, respectively) were conducted in two independent laboratories. To follow the regulation and law in Japan, these two experiments had to be conducted in the two separate laboratories.

Referee #2 (Remarks to the Author):

Strength

1. Though the phenomenon of forming larger and more numerous syncytia by B.1.1.7 alpha and B.1.351 beta variants has already been reported, this report is the first one to report the same phenomenon for B.1.617.2/Delta variant. See preprint in <https://www.biorxiv.org/content/10.1101/2021.06.11.448011v1>
2. More importantly, the authors have shown that the P681R caused more weight loss and higher lung viral load in the hamster model.
3. This study on the P681R mutation being the virulence factor of the B.1.617.2/Delta variant is timely and important.

Our reply:

We would like to thank the referee for understanding that our study is timely and important.

Revisions that can address the following weaknesses before publication:

1. The authors suggest that delta and P681R recombinant virus have an increased pathogenicity in hamsters. This claim need more support in terms of explaining:
 - a. How did the increased fusion and increased spike cleavage explain the decreased virus replication of delta (as shown in Fig 2c)?

Our reply:

According to the referee's comment (below, "Others, comment #3"), we performed multiple regression analyses including experimental conditions as explanatory variables and timepoints as qualitative control variables (please note that this analysis is theologically equivalent to two-way ANOVA, which was suggested by this referee). This statistical analysis showed that the replication kinetics of the Delta and B.1.1 is comparable ($P = 0.057$; **Fig. 2c** of the revised manuscript). We modified the text of the revised manuscript according to this result (page 6, lines 183-185).

- b. The only support for the increased pathogenicity comes from Fig 2d, which the authors showed a statistically significant difference between B.1.1 and delta on day 3 and day 4. As explained in the figure legend, there were 12 hamsters in the delta group. However, there is no variation between these 12 hamsters between day 1 to day 4. This is highly unusual. The

same problem applies to fig 4a. This would need further explanation and elaboration.

Our reply:

This (looking like few/no variation) would be because the data from hamsters (**Fig. 2c, 2d, and 4a-4c**) shown are the mean \pm S.E.M., not \pm S.D.

Regarding the error bars and how to show the variation – in a recent SARS-CoV-2-related paper published at *Nature* (Johnson et al., *Nature*, 2021. PMID 33494095), the authors adopted S.E.M. Therefore, we believe that showing these data by the mean with S.E.M. is acceptable. Additionally, we noticed that we did not explain the data and error bars shown in the figures at all. We apologize our insufficient explanation. In the revised manuscript, we explained them in the figure legend of the revised manuscript.

c. Based on the modest differences presented in fig2d and fig4a-b, the authors' conclusion on the increased pathogenicity of delta or P681 virus in hamsters need to be tuned down.

Our reply:

We added the data showing that the Delta variant is more pathogenic than the parental B.1.1 virus (**Fig. 2e-2h** of the revised manuscript). We think this is the first study directly showing the higher pathogenicity of the Delta variant *in vivo*.

We also showed the pathological data of the P681R-bearing virus (**Fig. 4d, 4e** of the revised manuscript). The difference by the P681R mutation may look smaller than the difference between the Delta variant and the parental B.1.1 virus (**Fig. 2e-2h** of the revised manuscript). Therefore, according to the comment from this referee, the explanations for the findings of the P681R-bearing virus were toned down (e.g., page 4, lines 111; page 11, lines 364-369; page 12, line 405).

To further address the referee's concern (please also see the reply to "Others, comment #3" below), we performed multiple regression analyses including experimental conditions as explanatory variables and timepoints as qualitative control variables. As shown in the revised manuscript, these statistical tests showed that the Delta and P681R-bearing viruses are significantly more pathogenic than the parental viruses (**Fig. 2c, 2d, 2f, 4a and 4b** of the revised manuscript). We believe that these additional data are robust and support our conclusion.

2. The authors suggest that the cell-cell fusion mediated by P681R results in an increased

resistance to neutralizing antibodies, which then leads to an increased pathogenicity. This claim may need more evidence to support:

a. There is no difference between D614G and D614G/P681R in the fusion experiment in the presence of neutralizing antibodies. The authors need to explain on this lack of difference.

b. Thus the authors need further experiment to support this claim, which is the key message of this study.

Our reply:

We agree with the reviewer's comments. This issue was not fully addressed in our study. Because these data (**Extended Data Fig. 8c** of the original manuscript) were not directly related to the conclusion of our current study (these results were not explained in the abstract of the original manuscript at all), we omitted these data from the revised manuscript.

3. At least in the syncytial assays, the alpha and beta variant should be included as controls.

Our reply:

Fig. R3. Syncytium formation.

Syncytia in infected VeroE6/TMPRSS2 cells were observed at 72 hpi. The size distributions of floating syncytia in the cultures infected with B.1.1 (n = 215), B.1.1.7/Alpha (n = 199), B.1.351/Beta (n = 249), B.1.617.2/Delta (n = 216), D614G (n = 228) and D614G/P681R (n = 164) viruses. The size distribution of the floating single cells in uninfected culture (n = 177) is also shown as a negative control. Statistically significant differences versus parental viruses (*, $P < 0.05$) or uninfected cells (#, $P < 0.05$) were determined by the Mann-Whitney U test.

To satisfy the referee's suggestion, we added the data showing syncytia formation using B.1.1, B.1.1.7/Alpha, B.1.351/Beta, B.1.617.2/Delta, D614G, and D614G/P681R viruses. As shown above (**Fig. R3**; this is a combined figure of **Fig. 2b** and **Fig. 3b** of the revised manuscript), the sizes of syncytia formed by the infections of B.1.617.2/Delta and D614G/P681R viruses are comparable, suggesting that the fusogenicity of the Delta variant

and the P681R-bearing virus is comparable. More importantly, the size of syncytia stimulated by B.1.617.2/Delta was significantly greater than that by B.1.1.7/Alpha and B.1.351/Beta (**Fig. R3**; also shown in **Fig. 2b** of the revised manuscript).

Additionally, in **Extended Data Fig. 5** of the revised manuscript, we showed that the fusogenicity of S proteins of all VOCs tested (i.e., B.1.1.7/Alpha, B.1.351/Beta, and B.1.617.2/Delta) was significantly greater than that of the parental D614G S. Moreover, the B.1.617.2/Delta S exhibited the highest fusogenicity with statistical significance. These results suggest that the B.1.617.2/Delta variant promotes syncytium formation more strongly than does the D614G-bearing B.1.1 virus as well as the B.1.1.7/Alpha and B.1.351/Beta VOCs. These additional data were explained in the revised manuscript (page 6, lines 167-178).

Others:

1. Extended Data Fig 2. At 72 hpi, there are one and three positive cells for B.1.1 and delta virus infected VeroE6/TMPRSS2 in the entire field, respectively. The number of infected cell is disproportionately low even when the cells are infected at 0.01 MOI.

Our reply:

We repeated this experiment, and the new data at 24 hpi were shown in **Extended Data Fig. 3** of the revised manuscript.

2. Extended Data Fig 8C. All three antibodies increased virus fusion by 100-200% at all dilutions. This is suggesting the presence of a potential systemic problem with the fusion assays that may infringe on the validity of these assays or the claims by the authors.

Our reply:

This is related to the comment 2 from this referee above. This concern was not fully addressed, and our data (**Extended Data Fig. 8c** of the original manuscript) were insufficient to conclude. Because this concern is not directly related to the conclusion of our current study, we omitted these data from the revised manuscript.

3. The correct statistical tests should be used instead of Student's t-tests. Two-way ANOVA should be used in Fig 2a, c, d, as well as for Fig 4a-c.

Our reply:

We appreciate the referee's very important suggestion. According to the reviewer's comment, we added multiple regression analyses including experimental conditions as explanatory variables and timepoints as qualitative control variables. This analysis is theologically equivalent to two-way ANOVA (**Fig. 2c, 2d, 4a 4b** of the revised manuscript). Additionally, we performed histopathological analysis in the lung of infected hamsters (**Fig. 2e** of the revised manuscript). We scored the pathological status, and the data were statistically analyzed by the multiple regression analyses with multiple testing correction by the Holm method (**Fig. 2f** of the revised manuscript). As shown in the revised manuscript, these statistical tests showed that the B.1.617.2/Delta and P681R-bearing viruses are significantly more pathogenic than the parental viruses. These new data are quite important to support the robustness of our findings. We appreciate the referee's great suggestion.

Regarding the statistical tests: we noticed that two-sided, unpaired Student's *t* test (without adjustments for multiple comparisons) was used in at least two recent SARS-CoV-2-related papers published at *Nature*:

Zhou et al., *Nature*, 2021. PMID 33636719

Johnson et al., *Nature*, 2021. PMID 33494095

Thanks for the referee's suggestion, now we showed the higher pathogenicity of the Delta variant and P681R-bearing virus than the parental viruses (**Fig. 2c, 2d, 2f, 4a, 4b** of the revised manuscript). However, we cannot say which time point(s) are specifically different by multiple regression analyses. Therefore, in addition to multiple regression analyses, we followed the statistical test used in the previous studies [two-sided, unpaired Student's *t* test (without adjustments for multiple comparisons)] to show the difference at each time point. To clarify the statistical tests applied, we mentioned the statistical tests used in these experiments in the revised manuscript.

Referee #3 (Remarks to the Author):

Saito et al. report spike mutation P681R enhances the fusogenicity and pathogenicity of Delta SARS-CoV-2. They used cell culture (Vero, Vero expressing TMPRSS, and Calu-3 cells), pseudovirus expressing SARS-CoV-2 spikes, cell-to-cell- fusion assay, and hamster models to analyze the P681R mutation. Overall, the results are interesting and supporting the conclusions. The following points should be addressed to substantiate the study.

Major comments

1. The cell-to-cell fusion experiment should include a control of no-hACE2 expression in target cells. This will examine if hACE2 is required for the cell-to-cell fusion.

Our reply:

Thanks for the important suggestion. To address the reviewer's concern, we showed that our cell-to-cell fusion assay is dependent of human ACE2 (**Extended Data Fig. 4** of the revised manuscript).

2. A more detailed virus entry experiment should be added to clearly demonstrate the potential role of P618R mutation in virus entry (or not). For example, the authors can compare the intracellular viral RNA levels between D614G and D614G/P681R viruses at 0.5 or 1 h post infection.

Our reply:

At 0.5-1 h postinfection, most of the viral RNA in the infected cells would be derived from the input virus attached/absorbed on the surface of target cells. Therefore, if we quantify the viral RNA level by real-time RT-PCR at these time points, these values would neither indicate the intracellular viral RNA levels nor reflect the efficacy of virus entry. If our understanding is correct, there are no established SARS-CoV-2 assays to particularly quantify/monitor the efficacy of virus entry. Rather, our cell-based fusion assay directly reflects the efficacy of virus entry mediated by SARS-CoV-2 S and human ACE2 (**Extended Data Fig. 4** of the revised manuscript). Moreover, we believe our data shown in **Fig. 3f-3h** of the revised manuscript are direct evidence showing that the P681R mutation plays a role in virus entry and are the answer to the referee's concern.

3. The authors should mention why hamster transmission experiments were not performed or presented.

Our reply:

As mentioned by the referee, virus transmission experiment was not the scope of this study, and therefore, we did not perform such experiments. We mentioned this reason in the revised manuscript (page 12, lines 407-414).

However, two recent cohort study published in *Lancet* (Sheikh et al.):

[https://www.thelancet.com/journals/lancet/article/PIIS0140-6736\(21\)01358-1/fulltext](https://www.thelancet.com/journals/lancet/article/PIIS0140-6736(21)01358-1/fulltext)

and *Lancet Infect Dis* (Twohig et al.):

[https://www.thelancet.com/journals/laninf/article/PIIS1473-3099\(21\)00475-8/fulltext](https://www.thelancet.com/journals/laninf/article/PIIS1473-3099(21)00475-8/fulltext)

suggested that patients with the Delta variant had more than two times the risk of hospital admission compared with patients with the Alpha variant. We believe that not only the higher transmissibility but also the (possible) higher pathogenicity of the Delta variant is an important and urgent concern. To clarify the risk of Delta variant infection and appeal the importance of our study, we referred these papers in the revised manuscript (references 16 and 17 of the revised manuscript).

4. The authors should discuss how these results are associated with the increased transmissibility of Delta variant observed in humans.

Our reply:

According to the suggestion, we mentioned this in the revised manuscript (pages 12-13, lines 420-428).

Reviewer Reports on the First Revision:

Referee #1 (Remarks to the Author):

The revised version of this manuscript appears improved and the authors appropriately addressed the reviewers' comments. The claim of increased pathogenicity is still not extraordinary well supported by the provided data. However, as expected differences are very small and the authors have toned down their statements. The reader is now able to judge the (small) differences because relevant data have been added. Overall, this is a timely and important contribution to our growing knowledge on SARS-CoV-2 VOCs.

Referee #2 (Remarks to the Author):

Sato et al has addressed all my queries satisfactorily.

Especially important is that they have demonstrated not just that the fusogenicity of S proteins of all VOCs tested (i.e., B.1.1.7/Alpha, B.1.351/Beta, and B.1.617.2/Delta) was significantly greater than that of the parental D614G S. But that the B.1.617.2/Delta S exhibited the highest fusogenicity with good statistical significance.

These results clearly suggest that the B.1.617.2/Delta variant promotes syncytium formation more strongly than does the D614G-bearing B.1.1 virus as well as the B.1.1.7/Alpha and B.1.351/Beta VOCs.

Referee #3 (Remarks to the Author):

Although the authors have addressed some points of this reviewer, they have not experimentally addressed the two most important comments of this reviewer, which are comments 2 & 3.
Does P681R affect virus entry and/or fusion?
Does Delta variant enhance transmission in the used hamster model?

Author Rebuttals to First Revision:

Referee #1 (Remarks to the Author):

The revised version of this manuscript appears improved and the authors appropriately addressed the reviewers' comments. The claim of increased pathogenicity is still not extraordinary well supported by the provided data. However, as expected differences are very small and the authors have toned down their statements. The reader is now able to judge the (small) differences because relevant data have been added. Overall, this is a timely and important contribution to our growing knowledge on SARS-CoV-2 VOCs.

Our reply:

We are happy to hear that our revised manuscript has satisfied the referee's concerns. We would like to thank the referee for understanding the importance of our study.

Referee #2 (Remarks to the Author):

Sato et al has addressed all my queries satisfactorily.

Especially important is that they have demonstrated not just that the fusogenicity of S proteins of all VOCs tested (i.e., B.1.1.7/Alpha, B.1.351/Beta, and B.1.617.2/Delta) was significantly greater than that of the parental D614G S. But that the B.1.617.2/Delta S exhibited the highest fusogenicity with good statistical significance.

These results clearly suggest that the B.1.617.2/Delta variant promotes syncytium formation more strongly than does the D614G-bearing B.1.1 virus as well as the B.1.1.7/Alpha and B.1.351/Beta VOCs.

Our reply:

We are happy to hear that our revised manuscript has satisfied the referee's queries. Thanks for the referee's important suggestion, we could improve our study. We appreciate the referee's helpful comments.

Referee #3 (Remarks to the Author):

Although the authors have addressed some points of this reviewer, they have not experimentally addressed the two most important comments of this reviewer, which are comments 2 & 3.

Does P681R affect virus entry and/or fusion?

Dese Delta variant enhance transmission in the used hamster model?

Our reply:

We are happy to hear that our revision could address some of the referee's concerns. We are sorry that we could not fully satisfy the referee's comments, but the following is our replies to the comments:

Does P681R affect virus entry and/or fusion?

Our reply: Yes. As shown in the manuscript, we clearly showed that the P681R mutation affects viral fusion (**Fig. 3f-3h**). We believe these data would be the answer to the referee's concern.

Dese Delta variant enhance transmission in the used hamster model?

Our reply: As mentioned in the revised manuscript, here we particularly focused on the pathogenicity of the Delta variant, and the transmissibility of this variant was not the main scope of this study.

In the comments 3 & 4, the referee has requested to "*mention why hamster transmission were not performed and presented*" and "*discuss how these results are associated with the increased transmissibility of Delta variant observed in human*". According to the referee's comments, we explained this issue in the Discussion section of revised manuscript. However, the referee has not requested to perform transmission experiments in the comments. Therefore, we think our revised manuscript has satisfied the referee's requests.

Cf. Comments from Referee #3 (Remarks to the Author) for our initial manuscript:

Saito et al. report spike mutation P681R enhances the fusogenicity and pathogenicity of Delta SARS-CoV-2. They used cell culture (Vero, Vero expressing TMPRSS, and Calu-3 cells), pseudovirus expressing SARS-CoV-2 spikes, cell-to-cell- fusion assay, and hamster models to analyze the P681R mutation. Overall, the results are interesting and supporting the conclusions. The following points should be addressed to substantiate the study.

Major comments

1. The cell-to-cell fusion experiment should include a control of no-hACE2 expression in target cells. This will examine if hACE2 is required for the cell-to-cell fusion.

Our reply:

Thanks for the important suggestion. To address the reviewer's concern, we showed that our cell-to-cell fusion assay is dependent of human ACE2 (**Extended Data Fig. 4** of the revised manuscript).

2. A more detailed virus entry experiment should be added to clearly demonstrate the potential role of P618R mutation in virus entry (or not). For example, the authors can compare the intracellular viral RNA levels between D614G and D614G/P681R viruses at 0.5 or 1 h post infection.

Our reply:

At 0.5-1 h postinfection, most of the viral RNA in the infected cells would be derived from the input virus attached/absorbed on the surface of target cells. Therefore, if we quantify the viral RNA level by real-time RT-PCR at these time points, these values would neither indicate the intracellular viral RNA levels nor reflect the efficacy of virus entry. If our understanding is correct, there are no established SARS-CoV-2 assays to particularly quantify/monitor the

efficacy of virus entry. Rather, our cell-based fusion assay directly reflects the efficacy of virus entry mediated by SARS-CoV-2 S and human ACE2 (**Extended Data Fig. 4** of the revised manuscript). Moreover, we believe our data shown in **Fig. 3f-3h** of the revised manuscript are direct evidence showing that the P681R mutation plays a role in virus entry and are the answer to the referee's concern.

3. The authors should mention why hamster transmission experiments were not performed or presented.

Our reply:

As mentioned by the referee, virus transmission experiment was not the scope of this study, and therefore, we did not perform such experiments. We mentioned this reason in the revised manuscript (page 12, lines 407-414).

However, two recent cohort study published in *Lancet* (Sheikh et al.):

[https://www.thelancet.com/journals/lancet/article/PIIS0140-6736\(21\)01358-1/fulltext](https://www.thelancet.com/journals/lancet/article/PIIS0140-6736(21)01358-1/fulltext)

and *Lancet Infect Dis* (Twohig et al.):

[https://www.thelancet.com/journals/laninf/article/PIIS1473-3099\(21\)00475-8/fulltext](https://www.thelancet.com/journals/laninf/article/PIIS1473-3099(21)00475-8/fulltext)

suggested that patients with the Delta variant had more than two times the risk of hospital admission compared with patients with the Alpha variant. We believe that not only the higher transmissibility but also the (possible) higher pathogenicity of the Delta variant is an important and urgent concern. To clarify the risk of Delta variant infection and appeal the importance of our study, we referred these papers in the revised manuscript (references 16 and 17 of the revised manuscript).

4. The authors should discuss how these results are associated with the increased transmissibility of Delta variant observed in humans.

Our reply:

According to the suggestion, we mentioned this in the revised manuscript (pages 12-13, lines 420-428).